# CGR: Confidence-Guided Replay for Buffer-Based Continual Learning

## Abstract

Continual Learning (CL) aims to acquire new knowledge while preserving previously learned information without catastrophic forgetting. Buffer-based methods, which retain samples from past tasks, have demonstrated promising results; however, efficiently allocating limited buffer space remains a significant challenge. Recent studies often either neglect the varying impact individual samples have on the learning process or incur high computational costs to identify informative replay samples. To overcome these limitations, we propose a novel approach called Confidence-Guided Replay (CGR), a lightweight **buffer** policy **for offline, task-aware continual supervised classification** that dynamically allocates the buffer by monitoring confidence fluctuations in the main continual learner model. Leveraging measures of sample contribution and difficulty, CGR adaptively prioritizes highly informative samples within the buffer, ~~significantly~~ enhancing knowledge retention and utilization efficiency. Our approach provides a flexible solution for dynamic buffer allocation, effectively addressing the varying importance and learning complexity of samples over time, and improves CL performance.[1]

## 1 Introduction

Continual Learning (CL), the gradual acquisition of new concepts (classes or tasks) without forsaking previous ones, stands as a pivotal capability in machine learning. Various methodologies, including regularization techniques (Kirkpatrick et al., 2017; Chaudhry et al., 2018a), architecture-based approaches (Mallya & Lazebnik, 2018; Hung et al., 2019), and rehearsal-based strategies (Chaudhry et al., 2018b; 2019; Aljundi et al., 2019b; Chaudhry et al., 2021; Sun et al., 2022; 2023; Tong et al., 2025), have been explored. However, the central challenge remains in striking a balance between assimilating new concepts (plasticity) and preserving existing knowledge (stability) (Kim et al., 2023; Kim & Han, 2023).

Although buffer-based methods outperform other approaches by mitigating catastrophic forgetting through the storage and reuse of samples from previously learned concepts in a small buffer, memory constraints make the buffer-filling strategy a critical challenge. While this approach can achieve strong performance (Tong et al., 2025), its effectiveness heavily depends on how the limited memory is utilized.

Some buffer-based methods, such as Chaudhry et al. (2019); Lin et al. (2023); Lopez-Paz & Ranzato (2017); Chaudhry et al. (2018b), do not use explicit buffer update policies. For example, Chaudhry et al. (2019); Lin et al. (2023) rely on reservoir sampling, while Lopez-Paz & Ranzato (2017); Chaudhry et al. (2018b) retain the earliest samples from each task. To improve buffer quality, other methods propose sample selection strategies based on gradient informativeness (Aljundi et al., 2019b; Jin et al., 2021; Yoon et al., 2021; Tiwari et al., 2022), or on Shapley values and influence functions (Shim et al., 2021; Sun et al., 2022; 2023). However, these approaches are typically computationally expensive and slow (see Table 6).

Moreover, prototype-based approaches such as iCaRL (Rebuffi et al., 2017), while gradient-free and sometimes effective in improving buffer utility, incur additional computational overhead to compute and update class feature means. Hardest-to-classify-sample strategy (Aljundi et al., 2019a) can be beneficial in certain

---

[1]Code is available in the supplementary materials.

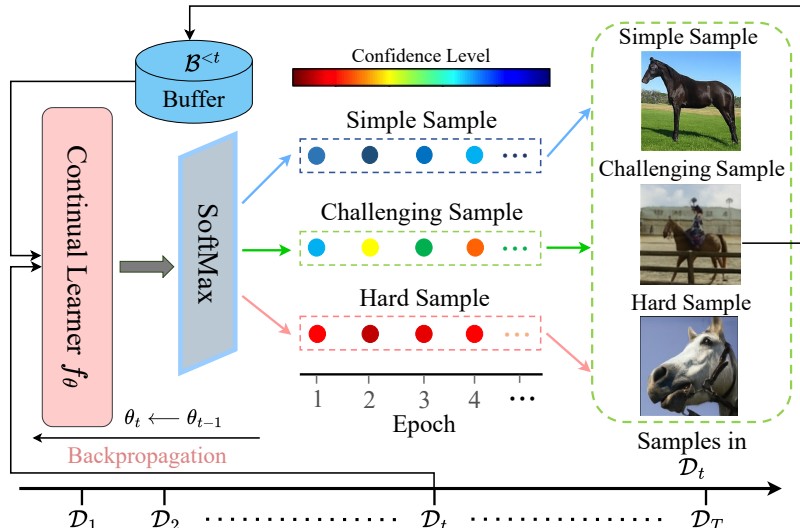

Figure 1: By tracking the target confidence of the main continual learner $f_\theta$ during training on each task $\mathcal{D}_t$, our method efficiently prioritizes replaying class samples with the highest confidence fluctuations (i.e., challenging, boundary samples), promoting robust decision boundaries and improving classification margins at minimal cost, unlike prior methods that rely on auxiliary models or gradient computations.

cases, but often risks allocating buffer space to outliers. Importantly, both types of methods tend to overlook samples whose confidence fluctuates over time—a hallmark of low-margin, decision-boundary-critical examples.

A recent approach, CSReL (Tong et al., 2025), avoids gradient computations by using forward-pass-only loss differences between a model trained on the coreset and another trained on the full task data. While more efficient than gradient- or influence-based methods, CSReL still requires training auxiliary models per task and performing forward passes per candidate samples, introducing non-trivial computational overhead (see Table 7). This added complexity can limit its practicality in resource-constrained settings. Moreover, because the auxiliary models are trained independently for each task and discarded after selection, they remain decoupled from the main continual learner and do not explicitly capture the informative dynamics of samples over training time—an aspect crucial for identifying boundary-critical or challenging examples.

To address the aforementioned challenges, we propose Confidence-Guided Replay (CGR) (Figure 1), a lightweight and effective method that leverages the confidence dynamics of the main continual learner, enabling a buffer update mechanism at minimal computational cost that does not require gradient computation, influence estimation, or auxiliary training—unlike prior methods—to identify and retain the most informative training samples (i.e., challenging, boundary samples). CGR tracks the variance in the target confidence for each sample across training epochs, targeting those with high variability—indicative of ambiguity and proximity to decision boundaries. CGR is modular and compatible with existing buffer-based methods, improving their performance without altering their core learning processes. Its design aligns with margin-theoretic principles, which emphasize the value of low-margin, uncertain samples for robust generalization (Bartlett & Mendelson, 2002). By continually reinforcing these high-variance examples, CGR enhances boundary stability and mitigates catastrophic forgetting.

Traditionally, samples with consistently high confidence are viewed as simple (prototypical), while those with persistently low confidence are seen as hard (outliers). In contrast, CGR identifies a critical third category: challenging samples—those with high variance in confidence over time. These are neither confidently learned nor consistently misclassified, but instead indicate decision boundary uncertainty. By prioritizing these samples, CGR selectively reinforces the most informative and challenging instances encountered during training.

The primary contributions of this work are:

1. We address the high computational cost associated with existing rehearsal-based CL methods that rely on complex buffer update policies.

2. We propose a lightweight buffer update strategy **for supervised classification-based CL with identifiable task boundaries** based solely on the confidence trajectory of the main model, without requiring auxiliary models, gradients, or **high** additional computations.

3. We demonstrate the effectiveness and efficiency of CGR through extensive experiments on standard ~~CL~~ **offline task-aware class- and task-incremental classification** benchmarks, and show its compatibility as a plug-in to boost the performance of existing CL methods.

## 2 Related Work

**Rehearsal-based CL Methods.** Rehearsal-based methods prevent catastrophic forgetting by maintaining a memory buffer of past data and periodically retraining the model with both old and new information. This approach enhances the model's ability to retain the knowledge of previous tasks (Chaudhry et al., 2018b; 2019; Aljundi et al., 2019b; Prabhu et al., 2020; Chaudhry et al., 2021; Sun et al., 2022; 2023; Tong et al., 2025).

A-GEM (Chaudhry et al., 2018b) leverages previous training data to minimize gradient interference explicitly. They populate the buffer using the earliest samples from each task, ensuring equal task-wise allocation. ER (Chaudhry et al., 2019) presents a straightforward rehearsal method that updates memories through reservoir sampling and employs random sampling during memory retrieval. Despite its simplicity, it remains a strong baseline. GSS (Aljundi et al., 2019b) approaches updating the memory buffer as a constrained optimization problem, to maximize the diversity of sample gradients within the buffer. GDUMB (Prabhu et al., 2020) operates by greedily accumulating new training samples into the buffer, which it subsequently uses to train a model from scratch for testing. HAL (Chaudhry et al., 2021) employs the old training samples, which are more susceptible to forgetting, as "anchors" to stabilize their predictions. This method enhances replay by adding an objective that minimizes the forgetting of crucial learned data points. MetaSP (Sun et al., 2022) utilizes the Pareto optimum of example influence on stability and plasticity, thereby guiding updates to the model and storage management. SOIF (Sun et al., 2023) leverages second-order influences to make more informed decisions about which samples to retain in the buffer. CSReL (Tong et al., 2025) introduces a gradient-free coreset selection method using reducible loss (ReL), computed via forward-pass-only loss differences. Its variants include CSReL-CL for per-task selection, CSReL-CL-Prv, which reduces task interference using memory, CSReL-RS for streaming data via reservoir sampling scaled by ReL, and ReL-cmb for knowledge distillation. For fair comparison, we use CSReL-CL-Prv and CSReL-ER (based on CSReL-RS) as baselines.

**Data-centric.** Data-centric approaches emphasize the quality of data over the complexity of models. Data-centric artificial intelligence involves techniques that aim to enhance datasets, thereby enabling the training of models with fewer data requirements (Motamedi et al., 2021; Mazumder et al., 2022). Ignoring the critical importance of data has led to inaccuracies, biases, and fairness issues in real-world applications (Mazumder et al., 2022). Previous research reveals that not all samples are equally informative; some contribute substantially more to the learning process than others. This is highlighted in studies such as those by Toneva et al. (2018); Swayamdipta et al. (2020), along with related research by Motamedi et al. (2021); Mazumder et al. (2022). For instance, Toneva et al. (2018) and Swayamdipta et al. (2020) leverage model confidence during training to cleanse the dataset. Specifically, Swayamdipta et al. (2020) categorizes the dataset into easy-to-learn, hard-to-learn, and ambiguous subsets, whereas Toneva et al. (2018) distinguishes between forgettable and unforgettable samples. These methods allow for the assessment of each sample's contribution during training. Building on these approaches, we introduce CGR to identify and retain the most informative (i.e., challenging, boundary) samples in the buffer.

Table 1: Methodological comparison of rehearsal-based CL methods. All characterization columns refer to the *buffer-update step*, not to training-time mechanisms. *Selection signal*: what drives the buffer update. *Grad. free*: does the buffer-update step avoid gradient, Hessian, or influence-function computations? (Methods may still use gradients elsewhere during training (e.g., A-GEM, HAL)). *No aux. model*: does the method avoid training an auxiliary model for the buffer-selection? (Methods may still use auxiliary models elsewhere (e.g., GDUMB)). *Class & task balance*: does the buffer policy enforce balance across classes and tasks (CV = 0 in Table 9)? *Selection cost*: qualitative cost of the buffer-update step, independent of training-time costs. Total empirical runtime including training-time overhead is reported in Tables 6 and 7. ✓/ ✗ indicate the property is satisfied / not satisfied.

| Method | Selection signal | Grad. free | No aux. model | Class & task balance | Selection cost |
|---|---|---|---|---|---|
| iCaRL (Rebuffi et al., 2017) | herding (class-mean prototype) | ✓ | ✓ | class | medium |
| A-GEM (Chaudhry et al., 2018b) | earliest-per-task | ✓ | ✓ | task | low |
| ER (Chaudhry et al., 2019) | reservoir (random) | ✓ | ✓ | near-uniform task | low |
| GSS (Aljundi et al., 2019b) | gradient-space diversity | ✗ | ✓ | none | high |
| GDUMB (Prabhu et al., 2020) | greedy class-balanced | ✓ | ✓ | class | low |
| HAL (Chaudhry et al., 2021) | FIFO ring buffer | ✓ | ✓ | near-uniform task | low |
| MetaSP (Sun et al., 2022) | influence function | ✗ | ✓ | none | high |
| SOIF (Sun et al., 2023) | 2nd-order influence function | ✗ | ✓ | none | high |
| CSReL-ER (Tong et al., 2025) | reducible loss + reservoir | ✓ | ✗ | near-uniform task | medium |
| CSReL-CL-Prv (Tong et al., 2025) | reducible loss + holdout | ✓ | ✗ | task | high |
| **CGR (Ours)** | target confidence variance | ✓ | ✓ | class & task | low |

## 3 Preliminaries

Consider a model $f_\theta$ parameterized by $\theta$, which is trained incrementally on a series of tasks $\{1, 2, \ldots, T\}$. At a specific time $t$, the model processes the task $\mathcal{D}_t = \{(x_i^t, y_i^t), \mathcal{C}^t\}_{i=1}^{N^t}$, where $x_i^t$ represents the input data, $y_i^t$ the corresponding label, $N^t$ the number of samples in $\mathcal{D}_t$, and $\mathcal{C}^t$ indicates the classes specific to task $\mathcal{D}_t$. **Consider a parameterized model** $f_\theta : \mathcal{X} \to \mathbb{R}^{|\mathcal{C}|}$, **where the input space is** $\mathcal{X} \subseteq \mathbb{R}^d$ **(e.g.,** $d = 3 \times 32 \times 32$ **for Split CIFAR-100),** $\mathcal{C} = \cup_{t=1}^T \mathcal{C}^t$ **is the full label space, and** $f_\theta$ **outputs class logits. The model is trained incrementally on a sequence of tasks** $\{D_1, \ldots, D_T\}$ **with disjoint class sets** $\mathcal{C}^t \cap \mathcal{C}^{t'} = \emptyset$ **for** $t \neq t'$. **Each task is given by** $D_t = \{(x_i^t, y_i^t)\}_{i=1}^{N^t}$ **with** $x_i^t \in \mathcal{X}$ **and** $y_i^t \in \mathcal{C}^t$. The buffer $\mathcal{B}$ is used to store a subset of past examples to mitigate catastrophic forgetting while having a limited capacity, denoted as $B_{\text{size}}$. For each task $\mathcal{D}_t$, the model's objective is to learn the new task while retaining knowledge from previous tasks. The model parameters are updated from $\theta_{t-1}$ to $\theta_t$ by minimizing the loss function over the combined dataset of the current task and the buffer:

$$\theta_t^* = \arg\min_{\theta_t} \mathbb{E}_{(x,y)\sim(\mathcal{D}_t \cup \mathcal{B})}\left[\ell(f_{\theta_t}(x), y)\right], \tag{1}$$

where $\ell$ represents the **standard softmax cross-entropy** loss function, $\ell(f_\theta(x), y) = -\log\left[\text{softmax}(f_\theta(x))\right]_y$, **and** $[\cdot]_y$ **denotes the probability assigned to the target class. This loss is smooth and bounded below by zero as a function of the logits. The expectation is approximated in practice by minibatch stochastic-gradient-based training for** $m$ **epochs, as shown in Algorithm 1.**

**Evaluation Settings: Class- vs. Task-Incremental Learning. We evaluate CGR under the two standard supervised CL protocols:** *task-incremental learning* **(TIL) and** *class-incremental learning* **(CIL). Both protocols share the training procedure described above—the model is trained on a sequence of tasks** $\{D_1, \ldots, D_T\}$ **with disjoint class sets** $\mathcal{C}^t$—**and differ only in the information available at test time. In TIL, the task identity** $t$ **is provided to the model at inference, so prediction is restricted to the classes of that task; the model only needs to distinguish among the** $|\mathcal{C}^t|$ **classes of the queried task. In CIL, no task identity is provided, and the model must predict over the union of all classes seen so far; the model must therefore distinguish across tasks as well as within them. CIL is the harder of the two settings and is typically the primary metric of interest, as it more closely reflects deployment scenarios where task**

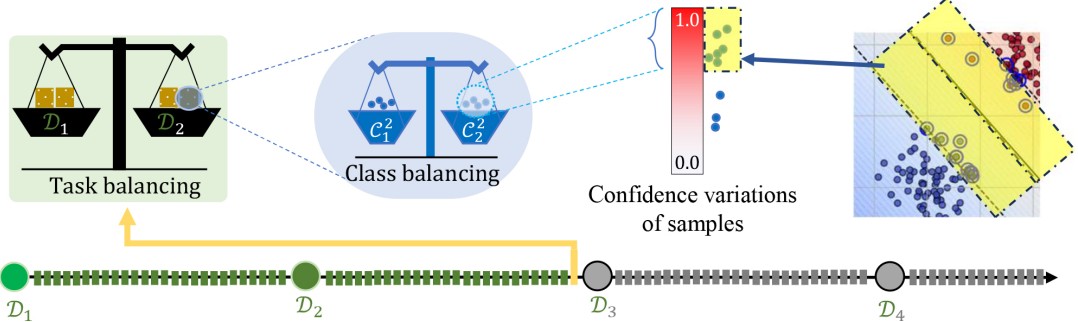

Figure 2: Illustration of the CGR buffer update policy. After training on each task $\mathcal{D}_t$, CGR updates the buffer by selecting the most challenging samples—those exhibiting high temporal variation in target class confidence—while maintaining both class- ($\mathcal{C}$) and task-level ($\mathcal{D}$) balance. The confidence variation heatmap highlights high-variance samples near decision boundaries (yellow region), which are critical for promoting robust generalization. By prioritizing these boundary-sensitive examples, CGR reinforces informative, low-margin instances that are most susceptible to forgetting.

boundaries are unavailable at test time. Following Boschini et al. (2022); Sun et al. (2023), we report ACC and BWT under both protocols on the same trained model: the only difference between Tables 2 and 3 is whether task identity is supplied to the classifier at evaluation, not how training is conducted.

## 4 Proposed Method: Confidence-Guided Replay (CGR)

In rehearsal-based CL, buffer updates play a key role in mitigating forgetting. Given the limited size of the buffer $\mathcal{B}$, the model must carefully decide which new samples to store and which old ones to discard. An effective strategy should retain the most valuable examples for future learning.

We propose Confidence-Guided Replay (CGR), a lightweight and modular buffer update policy designed to preserve boundary-critical examples—samples that lie near class decision boundaries and are therefore more susceptible to forgetting. Instead of relying on gradients, auxiliary models, or feature mean (prototype) calculation, CGR leverages a single, low-cost statistic derived from the main model itself: the temporal variance of the model's confidence in its target predictions for each training sample. This confidence trajectory—capturing how predicted target confidence fluctuates across initial training epochs—is used to identify and retain high-variance (i.e., boundary) samples. Specifically, CGR interprets high-variance samples as neither simple (consistently high confidence) nor hard outliers (consistently low confidence), but as challenging and informative decision-boundary samples. By explicitly retaining these ambiguous examples at the end of each task, CGR promotes decision boundary stability, reduces forgetting, and ensures a balanced, representative buffer through class-wise and task-wise selection (see Figures 1 and 2).

**Confidence Variance.** The target confidence of the model in its predictions is defined as the probability assigned to the target class $y_i^t$ for sample $x_i^t$ during task $\mathcal{D}_t$. Formally, the confidence score is expressed as:

$$\Gamma(x_i^t, y_i^t) = \cancel{\mathcal{P}(y_i^t \mid f_\theta(x_i^t))} \left[ \text{softmax}\left( f_\theta(x_i^t) \right) \right]_{y_i^t}, \tag{2}$$

~~where $f_\theta$ is the model that maps the input $x_i^t$ into class probabilities~~ **where $[\cdot]_{y_i^t}$ denotes the softmax probability assigned by the model $f_\theta$ to the target class $y_i^t$ given the input $x_i^t$.** During training, misclassified samples fall into two categories: boundary samples and outlier samples. Outlier samples consistently exhibit low confidence, whereas boundary samples display fluctuating confidence levels (Figure 1), reflecting uncertainty in their classification (Toneva et al., 2018; Swayamdipta et al., 2020).

---

**Algorithm 1** Confidence-Guided Replay (CGR)

---

1: **Input:** Task sequence $\{\mathcal{D}_1, \ldots, \mathcal{D}_T\}$, class counts $\{|\mathcal{C}^1|, \ldots, |\mathcal{C}^T|\}$, model $f_\theta$, buffer $\mathcal{B}$ with capacity $B_{\text{size}}$, CGR hyperparameter $E$, epochs number $m$, current-task and buffer batch size $b$

2: **Initialize:** Empty replay buffer $\mathcal{B} \leftarrow \emptyset$, parameters $\theta$

3: **for** $t = 1$ to $T$ **do**

4:      **for** epoch $e = 1$ to $m$ **do**

5:          **for** $(x_i^t, y_i^t)^b \sim \mathcal{D}_t$ **do**

6:              $(x_i^{\mathcal{B}}, y_i^{\mathcal{B}})^b \leftarrow \text{RandomRetrieval}(\mathcal{B})$

7:              $(\mathbf{x}, y) \leftarrow \big\{(x_i^t, y_i^t)^b, (x_i^{\mathcal{B}}, y_i^{\mathcal{B}})^b\big\}$

8:              **if** $e \leq E$ **then**

9:                  ~~$\Gamma_e(x_i^t, y_i^t)^b \leftarrow \mathcal{P}\big((y_i^t)^b \mid f_\theta((x_i^t)^b)\big)$~~ $\Gamma_e(x_i^t, y_i^t)^b \leftarrow \Big[\text{softmax}\big(f_\theta((x_i^t)^b)\big)\Big]_{(y_i^t)^b}$

10:              **end if**

11:              ~~$\theta_t^* = \arg\min_\theta \text{CrossEntropy}(f_\theta(\mathbf{x}), y)$~~      $\theta \leftarrow \theta - \eta\, \nabla_\theta \mathbf{CrossEntropy}\big(f_\theta(\mathbf{x}), y\big)$

12:          **end for**

13:      **end for**

14:      $\sigma^2(x^t, y^t) = \frac{1}{E} \sum_{e=1}^{E} \big[\Gamma_e(x^t, y^t) - \overline{\Gamma}(x^t, y^t)\big]^2$

15:      **if** $t \geq 2$ **then**

16:          $k_{\text{prune}} \leftarrow \frac{B_{\text{size}}}{(t-1)\,|\mathcal{C}^{t-1}|} - \frac{B_{\text{size}}}{t\,|\mathcal{C}^t|}$

17:          $\mathcal{B} \leftarrow \mathcal{B} \setminus \big\{(x_{i,c}^{<t}, y_{i,c}^{<t}) \,\big|\, \sigma^2(x_{i,c}^{<t}, y_{i,c}^{<t}) \in \text{bottom-}k_{\text{prune}}\big\}$

18:      **end if**

19:      $k_{\text{add}} \leftarrow \frac{B_{\text{size}}}{t\,|\mathcal{C}^t|}$

20:      $\mathcal{B} \leftarrow \mathcal{B} \cup \big\{(x_{i,c}^t, y_{i,c}^t) \,\big|\, \sigma^2(x_{i,c}^t, y_{i,c}^t) \in \text{top-}k_{\text{add}}\big\}$

21: **end for**

22: **Output:** Updated parameters $\theta$, and buffer $\mathcal{B}$

---

This fluctuation can be quantified through the variance of the model's confidence scores computed over the first $E$ epochs of training:

$$\sigma^2(x_i^t, y_i^t) = \frac{1}{E} \sum_{e=1}^{E} \big[\Gamma_e(x_i^t, y_i^t) - \overline{\Gamma}(x_i^t, y_i^t)\big]^2, \tag{3}$$

where the mean confidence is given by

$$\overline{\Gamma}(x_i^t, y_i^t) = \frac{1}{E} \sum_{e=1}^{E} \Gamma_e(x_i^t, y_i^t). \tag{4}$$

Here, $\Gamma_e(x_i^t, y_i^t)$ denotes the model's target confidence for sample $(x_i^t, y_i^t)$ at epoch $e$, and $\overline{\Gamma}(x_i^t, y_i^t)$ represents the average confidence across $E$ epochs. A higher value of $\sigma^2(x_i^t, y_i^t)$ indicates greater fluctuation in confidence, suggesting that the sample lies near a decision boundary where the model's predictions are less stable.

The full process of CGR is summarized in Algorithm 1, which illustrates how confidence-guided sample selection is integrated into the learning process.

**Buffer Structure.** To avoid long-tailed memory distributions that distort decision boundaries (Cui et al., 2019; Samuel & Chechik, 2021; Shi et al., 2023), CGR enforces a balanced allocation across both tasks and classes (Figure 2). This design mitigates over-representation of dominant categories and ensures uniform replay coverage.

Assuming that all tasks $\{\mathcal{D}_1, \ldots, \mathcal{D}_T\}$ contain an equal number of classes and samples, the replay buffer at time $t$ allocates an equal memory share to each task, i.e., $\frac{B_{\text{size}}}{t}$. For a task $\mathcal{D}_t$ with $|\mathcal{C}^t|$ classes, each class receives $\frac{B_{\text{size}}}{t \times |\mathcal{C}^t|}$ capacity. Under this assumption, maintaining class balance inherently ensures task balance.

Therefore, CGR does not require explicit task identifiers—only samples and class labels, as in conventional replay-based methods.

During buffer updates, samples with the highest variance within each class are retained, while those with the lowest variance are discarded. Because samples in each class-specific buffer segment are stored in descending order of variance relative to their buffer index, the variance values need not be stored explicitly. Thus, the buffer maintains only the samples and their corresponding labels.

**Adaptive Buffer Update and Pruning.** To accommodate memory constraints, CGR adopts an *adaptive buffer management strategy* that dynamically prunes less informative samples prior to integrating data from a new task $\mathcal{D}_t$. Specifically, for each previously observed class $c \in \mathcal{C}^{<t}$, the algorithm removes the $k_{\text{prune}}$ samples exhibiting the lowest variance from the replay buffer $\mathcal{B}$. The number of samples to be pruned per class is defined as:

$$k_{\text{prune}} = \frac{B_{\text{size}}}{(t-1)\left|\mathcal{C}^{t-1}\right|} - \frac{B_{\text{size}}}{t\left|\mathcal{C}^t\right|}. \tag{5}$$

Accordingly, the pruning operation is expressed as:

$$\mathcal{B} \leftarrow \mathcal{B} \setminus \left\{ (x_{i,c}^{<t}, y_{i,c}^{<t}) \,\middle|\, \sigma^2(x_{i,c}^{<t}, y_{i,c}^{<t}) \in \text{bottom-}k_{\text{prune}} \right\}. \tag{6}$$

This selective removal ensures that the buffer retains high-variance samples, which are generally more informative for preserving discriminative decision boundaries and mitigating catastrophic forgetting.

After pruning, the buffer is updated to incorporate samples from the current task $\mathcal{D}_t$. For each class $c \in \mathcal{C}^t$, the $k_{\text{add}}$ samples with the highest variance are selected and inserted into the available buffer space:

$$\mathcal{B} \leftarrow \mathcal{B} \cup \left\{ (x_{i,c}^t, y_{i,c}^t) \,\middle|\, \sigma^2(x_{i,c}^t, y_{i,c}^t) \in \text{top-}k_{\text{add}} \right\}, \tag{7}$$

where

$$k_{\text{add}} = \frac{B_{\text{size}}}{t\left|\mathcal{C}^t\right|}. \tag{8}$$

This formulation guarantees *class- and task-balanced memory allocation* while prioritizing the most informative and challenging samples, thereby enhancing the model's capacity for continual adaptation and stable performance across tasks.

**Scope and Assumptions. CGR is designed for supervised classification-based CL with identifiable task boundaries. The method assumes access to labeled examples from the current task for multiple training epochs, since it estimates sample informativeness from the temporal variance of target-class confidence during the first $E$ epochs. Therefore, CGR is directly applicable to task-incremental and class-incremental supervised classification settings, such as Split CIFAR-100, Split Mini-ImageNet, and Split Tiny-ImageNet. In its current form, CGR is not intended for strict single-pass online CL, task-free CL, unsupervised CL, reinforcement learning, regression, or settings where task boundaries are unavailable. Extending confidence-guided replay to these broader settings is an interesting direction for future work.**

### 4.1 Which Boundary?

As decision boundaries evolve during training, an essential question arises: which boundary should guide sample selection for buffer retention? Should we rely on boundaries influenced by outliers, or prioritize those shaped by challenging, informative samples? Data-centric studies (Toneva et al., 2018; Swayamdipta et al., 2020) suggest that early-stage uncertainty is a strong indicator of a sample's long-term contribution to learning.

To investigate this, we analyze CIFAR-10 in a batch training setup—analogous to a single task in Split CIFAR-100. For each sample, we track its target confidence across 4 and 50 epochs and visualize the mean and ~~variance~~ **standard deviation** in Figure 3, sub-figures (b.1) and (b.2). These plots reveal how confidence variability highlights boundary-critical samples early in training.

Figure 3: (a) The first, second, and third rows show simple, challenging, and hard samples, respectively, from five classes of CIFAR-10. (b) Visualization of the mean and ~~variance~~ **standard deviation** of target confidence for CIFAR-10 samples through epochs 4 and 50. (c) Performance comparison for different values of $E$ in our boundary sample identification, using a buffer size of 1000. CIL and TIL indicate class and task incremental settings, respectively.

We categorize samples observed in the first $E = 4$ epochs into three groups. *Area 1* (Simple): High confidence, low variance—well-classified and central. *Area 2* (Challenging): Moderate confidence, high variance—near boundaries and most informative. *Area 3* (Hard): Low confidence, low variance—rare or atypical, offering little value for boundary refinement.

Sub-figure (a) illustrates representative examples from these areas across five CIFAR-10 classes. Simple samples are easily identifiable, showcasing prominent class attributes or centrally-placed objects (e.g., a horse on grass). Challenging samples exhibit ambiguous traits, such as an airplane against a gray background, deviating from typical class patterns. Hard samples contain rare or atypical features—for instance, a horse showing only its head—and often act as outliers, diverging from the standard sample distribution.

By epoch 50, simple and previously challenging samples consolidate into *Area 4*, while hard samples and remaining challenging cases form *Area 5*. Comparing *Area 2* (early) to *Area 5* (late), we argue that retaining high variance (boundary) samples from early training better supports generalization and decision boundary robustness than late training.

Finally, sub-figures (c.1) and (c.2) assess sensitivity to the epoch window $E$. We find CGR remains effective for $E = 2$ to 6, beyond which performance declines as hard samples (outliers) transition to boundary (high variance) samples, and challenging to simple ones, reducing their impact on refining the decision boundary.

## 5 Experiments

**In this section, we evaluate CGR empirically. Section 5.1 describes the datasets, evaluation metrics, baselines, and implementation details. Section 5.2 reports the main class- and task-incremental results, the modularity of CGR as a plug-in to existing rehearsal methods, comparisons against alternative buffer-update policies, and runtime analyses. Section 5.3 presents ablation studies, including high-resolution evaluation, buffer-composition analysis, and hyperparameter sensitivity.**

### 5.1 Setup

For our experiments, we extended the Mammoth framework as described in Buzzega et al. (2020); Boschini et al. (2022) to ensure a fair setup for comparing CL methods. All experiments are conducted using a Quadro RTX 8000 equipped with CUDA Version 10.2.

**Datasets.** Our experiments utilized three benchmarks. The first, Split CIFAR-100, consists of 100 classes split into 10 tasks, each with 10 classes and $32 \times 32$ pixel images (Boschini et al., 2022). The second, Split Mini-ImageNet, divides 100 classes into 5 tasks of 20 classes each, with similarly resized images (Sun et al., 2022) (see the ablation study in Section 5.3 for results using the original $84 \times 84$ resolution). The third, Split

Tiny-ImageNet, includes 200 classes split into 10 tasks of 20 classes each, with images resized to $32 \times 32$ pixels (Buzzega et al., 2020). Each dataset has 500 training samples per class, with test samples numbering 100 for Split CIFAR-100 and Mini-ImageNet, and 50 for Split Tiny-ImageNet.

**Metrics.**   We employ two primary metrics to evaluate the performance of CL methods: Average Accuracy (ACC) and Backward Transfer (BWT), as outlined in Lopez-Paz & Ranzato (2017). The ACC metric assesses overall performance by calculating the average accuracy across all tasks after the model has completed training. Conversely, BWT measures the degree of knowledge loss over time by measuring the average forgetting across all previous tasks once training on all tasks has concluded. These metrics are mathematically defined as ACC $= \frac{1}{T} \sum_{j=1}^{T} \alpha_{T,j}$,   BWT $= \frac{1}{T-1} \sum_{j=1}^{T-1} \beta_{T,j}$. Here, $\alpha_{i,j}$ represents the accuracy on the test set held out for task $j$ after the network has been trained on task $i$. $\beta_{i,j}$ is calculated as $(\alpha_{i,j} - \alpha_{j,j})$, signifying the decrease in model performance on task $j$ due to training on subsequent tasks. $T$ shows the number of tasks. ~~We evaluate these metrics under class incremental learning (CIL) and task incremental learning (TIL) settings, as discussed in previous studies Boschini et al. (2022); Sun et al. (2023).~~ **We evaluate ACC and BWT under both the CIL and TIL protocols defined in Section 3.**

**Baselines.**   We compared our method against nine rehearsal-based approaches, including A-GEM (Chaudhry et al., 2018b), ER (Chaudhry et al., 2019), GSS (Aljundi et al., 2019b), GDUMB (Prabhu et al., 2020), HAL (Chaudhry et al., 2021), MetaSP (Sun et al., 2022), SOIF (Sun et al., 2023), and two variants of CSReL (Tong et al., 2025), CSReL-ER, and CSReL-CL-Prv.

**Implementation Details.**   For the implementation and hyperparameters of the baselines, we followed the Mammoth framework (Buzzega et al., 2020; Boschini et al., 2022), MetaSP (Sun et al., 2022), SOIF (Sun et al., 2023), and CSReL (Tong et al., 2025), using ResNet18 (He et al., 2016) as the backbone model, trained from scratch with cross-entropy loss. We ensured fairness by averaging results over five runs with fixed seeds (0 to 4). **To quantify the confidence of our comparisons, we report mean $\pm$ standard deviation over 5 seeds in Tables 2 and 3, and perform paired two-sided t-tests ($\alpha = 0.05$) between CGR and the strongest competitor, CSReL-CL-Prv, on matched (dataset, buffer, seed) runs. Full t-statistics, p-values, and decisions appear in Tables 11 and 12 (Appendix A.2). A \* next to a value in Tables 2 and 3 indicates that the method wins the cell significantly ($p < 0.05$, paired two-sided t-test).** Both the current and memory batch sizes were set to 32. ~~The hyperparameter $E$ was empirically set based on buffer size and dataset: for Split CIFAR-100, values of 3, 4, and 5 were used for buffer sizes 500, 1000, and 2000, respectively; for Split Mini-ImageNet, the values were 3, 3, and 4; and for Split Tiny-ImageNet, 3 was used across all buffer sizes. A sensitivity analysis of $E$ is provided in Ablation.~~ **For CGR, the only method-specific hyperparameter is $E$, the number of early epochs over which target-confidence variance is computed. We searched $E$ over the range $\{2, 3, \ldots, 10\}$ for each (dataset, buffer-size) pair. The selected values are $(3, 4, 5)$ for Split CIFAR-100 at buffer sizes $(500, 1000, 2000)$; $(3, 3, 4)$ for Split Mini-ImageNet; and $(3, 3, 3)$ for Split Tiny-ImageNet. As shown in the sensitivity analysis in Figure 3 (sub-figures c.1, c.2), CGR's performance is stable across $E \in [2, 6]$ on all three datasets.** We used the SGD optimizer for 50 epochs per task, with learning rates of 0.1 for Split CIFAR-100 and 0.03 for both Split Mini-ImageNet and Split Tiny-ImageNet, applying standard augmentations, including random cropping and random horizontal flipping (see Appendix A.1 for more details).

## 5.2   Main Results

**Class Incremental Learning.**   Table 2 presents the Average Accuracy (ACC) and Backward Transfer (BWT) results in the class incremental settings. ~~As is evident, our method consistently outperforms other methods in all scenarios for ACC. Regarding BWT, for Split CIFAR-100, we outperform others on average, but for two other datasets, we rank second. However, a closer analysis is warranted. In these cases, the HAL method achieves the best average BWT. Nonetheless, as results show, HAL has the worst average ACC, indicating that it failed to acquire sufficient knowledge to retain, much less forget, which raises concerns about its overall learning capability. Therefore, a fair comparison should consider both BWT and ACC, as a less negative BWT alone is insufficient if the ACC is low. To this end, we should consider BWT alongside ACC.~~

Table 2: Results of class-incremental learning: Average Accuracy (ACC, higher is better) and Backward Transfer (BWT, favoring less negative values), averaged over 5 runs. 'Mean' columns within each dataset show averages across buffer sizes, while the 'Mean' column next to the datasets indicates the overall average for each method across all datasets. The best results are bolded, and the second best are underlined. **\* indicates statistically significant improvement between CGR and the strongest competing baseline (CSReL-CL-Prv), using a two-sided paired t-test at $\alpha = 0.05$.** Note: BWT of the GDUMB method was excluded due to computational constraints.

| Method | Dataset | Split CIFAR-100 | | | | Split Mini-ImageNet | | | | Split Tiny-ImageNet | | | | Mean |
|---|---|---|---|---|---|---|---|---|---|---|---|---|---|---|
| | Buffer | 500 | 1000 | 2000 | Mean | 500 | 1000 | 2000 | Mean | 500 | 1000 | 2000 | Mean | |
| A-GEM | ACC | 9.32±0.11 | 9.23±0.04 | 9.25±0.13 | 9.27±0.07 | 14.65±0.17 | 14.69±0.32 | 14.74±0.08 | 14.69±0.11 | 6.30±0.13 | 6.30±0.37 | 6.23±0.27 | 6.28±0.16 | 10.08±0.08 |
| | BWT | -85.68±1.15 | -85.05±1.38 | -86.07±0.65 | -85.60±0.65 | -66.56±0.84 | -66.79±1.02 | -66.60±0.75 | -66.65±0.85 | -64.18±1.09 | -64.71±0.93 | -64.61±0.67 | -64.50±0.75 | -72.25±0.63 |
| ER | ACC | 19.29±0.39 | 24.46±0.84 | 31.94±1.26 | 25.23±0.56 | 18.02±0.26 | 20.78±0.28 | 24.72±0.59 | 21.17±0.31 | 8.90±0.40 | 10.75±0.10 | 13.88±0.37 | 11.18±0.14 | 19.19±0.28 |
| | BWT | -75.53±0.49 | -69.29±0.88 | -61.18±1.67 | -68.67±0.65 | -63.62±0.39 | -60.07±0.60 | -55.12±0.66 | -59.60±0.22 | -68.53±0.57 | -67.12±0.51 | -63.47±0.69 | -66.37±0.38 | -64.88±0.31 |
| GSS | ACC | 22.23±0.59 | 28.84±0.68 | 35.27±0.86 | 28.78±0.23 | 18.48±0.48 | 21.91±0.47 | 28.16±0.28 | 22.85±0.35 | 9.23±0.23 | 11.45±0.25 | 15.52±0.41 | 12.07±0.18 | 21.23±0.24 |
| | BWT | -70.26±1.13 | -61.81±0.47 | -53.51±1.11 | -61.86±0.31 | -61.72±0.34 | -55.95±0.99 | -46.80±0.51 | -54.82±0.39 | -67.56±0.33 | -64.05±0.32 | -58.06±0.35 | -63.22±0.17 | -59.97±0.23 |
| GDUMB | ACC | 10.36±0.56 | 16.35±0.12 | 25.47±0.52 | 17.39±0.26 | 7.09±0.73 | 10.15±0.53 | 14.33±0.29 | 10.52±0.36 | 3.59±0.26 | 5.15±0.33 | 7.53±0.19 | 5.42±0.22 | 11.11±0.17 |
| | BWT | – | – | – | – | – | – | – | – | – | – | – | – | – |
| HAL | ACC | 10.23±1.23 | 12.62±0.81 | 16.68±1.84 | 13.18±0.67 | 5.95±0.72 | 7.10±1.08 | 8.59±1.42 | 7.21±0.45 | 2.45±0.21 | 2.58±0.39 | 2.91±0.26 | 2.65±0.19 | 7.68±0.17 |
| | BWT | **-55.71±0.79** | -53.55±2.34 | -47.52±2.72 | -52.26±1.29 | **-38.88±1.62** | **-38.11±1.65** | **-36.63±2.56** | **-37.87±1.47** | **-34.95±1.72** | **-34.65±1.56** | **-32.17±1.78** | **-33.92±0.93** | **-41.35±0.75** |
| MetaSP | ACC | 21.87±0.97 | 26.23±0.77 | 29.74±1.77 | 25.95±0.96 | 22.92±0.48 | 27.84±0.90 | 33.39±0.45 | 28.05±0.46 | 11.34±0.37 | 14.72±0.33 | 19.22±0.53 | 15.09±0.20 | 23.03±0.38 |
| | BWT | -71.01±1.24 | -64.55±1.44 | -59.90±2.36 | -65.48±1.29 | -60.40±1.08 | -53.28±1.32 | -44.64±0.62 | -52.77±0.82 | -67.63±0.60 | -63.03±0.47 | -56.61±0.60 | -62.42±0.39 | -60.23±0.71 |
| SOIF | ACC | 18.85±1.02 | 26.19±0.62 | 25.36±0.91 | 23.47±0.71 | 18.75±0.32 | 22.85±0.86 | 23.46±0.49 | 21.69±0.45 | 9.10±0.18 | 11.66±0.23 | 11.58±0.24 | 10.78±0.03 | 18.64±0.29 |
| | BWT | -71.47±1.12 | -62.91±0.83 | -62.11±0.96 | -65.50±0.72 | -65.70±0.25 | -58.65±1.18 | -54.76±0.48 | -59.70±0.53 | -69.63±0.48 | -65.65±0.60 | -63.64±0.37 | -66.31±0.32 | -63.84±0.35 |
| CSReL-ER | ACC | 20.29±0.57 | 27.65±0.55 | 36.37±0.58 | 28.10±0.44 | 19.11±0.31 | 23.54±0.28 | 29.22±0.58 | 23.96±0.33 | 9.70±0.28 | 12.10±0.44 | 16.58±0.27 | 12.79±0.28 | 21.62±0.15 |
| | BWT | -74.02±0.54 | -65.41±0.53 | -55.05±0.94 | -64.83±0.50 | -62.96±0.62 | -56.74±0.33 | -50.52±0.62 | -56.74±0.42 | -68.22±0.49 | -65.38±0.49 | -59.71±0.32 | -64.44±0.37 | -62.00±0.14 |
| CSReL-CL-Prv | ACC | 26.61±0.27 | 32.27±2.64 | 35.18±3.84 | 31.35±1.06 | 23.31±0.71 | 28.25±0.94 | 34.54±0.86 | 28.70±0.23 | 11.50±0.51 | 15.57±0.12 | 21.15±0.56 | 16.07±0.18 | 25.38±0.36 |
| | BWT | -61.31±1.49 | -51.37±2.68 | -44.39±3.81 | -52.36±0.99 | -57.60±1.02 | -48.51±2.11 | -36.68±1.37 | -47.60±0.69 | -65.44±0.58 | -60.05±0.69 | -50.83±1.06 | -58.77±0.60 | -52.91±0.54 |
| **CGR** | ACC | **28.41±0.86\*** | **34.65±1.27** | **41.03±0.60\*** | **34.70±0.31\*** | **24.20±0.43\*** | **30.05±0.66\*** | **36.36±0.60\*** | **30.20±0.50\*** | **12.35±0.25\*** | **17.32±0.30\*** | **23.50±0.52\*** | **17.72±0.19\*** | **27.54±0.18\*** |
| | BWT | -60.03±2.33 | **-51.34±2.00** | **-41.53±3.36** | **-50.97±2.27** | -56.39±0.71 | -47.38±0.59 | -37.17±0.67 | -46.98±0.56 | -64.99±0.64 | -58.59±0.55\* | -49.42±0.70 | -57.67±0.56\* | -51.87±0.56\* |

~~Notably, although CSReL-CL-Prv incurs higher computational costs (see Table 7), our method consistently outperforms it.~~ **In terms of ACC, CGR ranks first in almost all CIL scenarios; paired two-sided t-tests against the strongest competitor, CSReL-CL-Prv, indicate that this advantage is statistically significant ($p < 0.05$) in 12 of 13 cells (Table 11). For BWT, CGR attains the best mean on Split CIFAR-100 and the second-best mean on the other two datasets; however, the advantage over CSReL-CL-Prv is individually significant in only 3 of 13 cells, while the pooled comparison across all datasets and buffer sizes is significant ($p = 0.010$). We therefore interpret CGR's BWT improvement as consistent in direction but modest in magnitude relative to run-to-run variability. On Split Mini- and Tiny-ImageNet, HAL attains a better mean BWT than CGR, but its ACC is the worst among all methods, indicating that it fails to acquire sufficient knowledge to retain. A fair comparison must therefore consider BWT alongside ACC, since a less negative BWT is uninformative when accuracy is low. Finally, while CSReL-CL-Prv is CGR's closest competitor in terms of accuracy, it incurs substantially higher computational cost (see Table 7).**

**Task Incremental Learning.** Table 3 shows results in the task incremental settings. ~~As is evident, for Split CIFAR-100, our method outperforms all others in both ACC and BWT. For the other two datasets, we rank second and perform comparably to CSReL-CL-Prv. On average, across all scenarios, we surpass CSReL-CL-Prv in ACC and achieve a BWT that is nearly identical. However, as shown in Table 7, CSReL-CL-Prv incurs substantial computational costs, limiting its practicality in resource-constrained settings.~~ **On Split CIFAR-100, CGR attains the best in both ACC and BWT across all buffer sizes, and its ACC advantage over CSReL-CL-Prv is statistically significant in all three cells (Table 12). On Split Mini-ImageNet and Split Tiny-ImageNet, the two methods trade places: CSReL-CL-Prv achieves the best ACC and BWT in all cells, with several of these advantages reaching statistical significance ($p < 0.05$). Averaged across all datasets and buffer sizes, CGR has a marginally higher ACC (66.52 vs. 66.06) and a nearly identical BWT (-11.02 vs. -10.95); neither of these overall differences is statistically significant in the 45-run paired comparison. We therefore characterize CGR and CSReL-CL-Prv as comparable in TIL performance, with CGR favored on Split CIFAR-100 and CSReL-CL-Prv favored on Split Mini- and Tiny-ImageNet. The practical advantage of CGR in the TIL setting is efficiency rather than accuracy: as shown**

Table 3: Results of task-incremental learning: Average Accuracy (ACC, higher is better) and Backward Transfer (BWT, favoring less negative values), averaged over 5 runs. 'Mean' columns within each dataset show averages across buffer sizes, while the 'Mean' column next to the datasets indicates the overall average for each method across all datasets. The best results are bolded, and the second best are underlined. **\* indicates statistically significant improvement between CGR and the strongest competing baseline (CSReL-CL-Prv), using a two-sided paired t-test at $\alpha = 0.05$.** Note: BWT of the GDUMB method was excluded due to computational constraints.

| Method | Dataset | Split CIFAR-100 | | | | Split Mini-ImageNet | | | | Split Tiny-ImageNet | | | | Mean |
|---|---|---|---|---|---|---|---|---|---|---|---|---|---|---|
| | Buffer | 500 | 1000 | 2000 | Mean | 500 | 1000 | 2000 | Mean | 500 | 1000 | 2000 | Mean | |
| A-GEM | ACC | 54.13±2.64 | 54.46±1.03 | 55.86±0.71 | 54.82±0.69 | 32.48±1.43 | 32.56±1.57 | 33.82±0.68 | 32.95±1.11 | 19.24±0.85 | 19.84±0.81 | 19.53±1.05 | 19.54±0.69 | 35.77±0.70 |
| | BWT | -35.89±3.05 | -34.78±2.17 | -34.29±1.23 | -34.99±0.75 | -44.26±2.74 | -44.46±2.65 | -42.74±1.56 | -43.82±2.24 | -49.80±1.45 | -49.67±0.90 | -49.84±1.49 | -49.77±1.06 | -42.86±1.27 |
| ER | ACC | 73.90±0.76 | 78.12±0.32 | 81.69±0.63 | 77.90±0.45 | 50.70±0.44 | 55.25±0.80 | 58.91±0.45 | 54.95±0.33 | 42.15±1.03 | 48.90±0.49 | 54.18±0.56 | 48.41±0.59 | 60.42±0.30 |
| | BWT | -15.40±0.44 | -10.70±0.84 | -6.97±0.98 | -11.02±0.66 | -22.92±0.72 | -17.42±1.14 | -13.15±0.22 | -17.83±0.51 | -31.73±0.94 | -25.16±0.66 | -19.42±1.09 | -25.44±0.72 | -18.10±0.21 |
| GSS | ACC | 72.85±0.28 | 76.75±1.82 | 80.89±0.57 | 76.83±0.63 | 50.23±0.32 | 54.68±0.64 | 59.38±0.33 | 54.76±0.28 | 40.33±1.93 | 47.20±0.58 | 52.80±0.58 | 46.78±0.80 | 59.46±0.51 |
| | BWT | -15.91±0.80 | -12.38±1.81 | -7.91±0.69 | -12.07±0.65 | -23.16±0.71 | -17.52±1.15 | -12.36±0.50 | -17.68±0.54 | -33.74±2.19 | -25.99±0.44 | -20.18±0.41 | -26.64±0.74 | -18.80±0.54 |
| GDUMB | ACC | 32.91±0.93 | 42.73±0.75 | 54.49±0.82 | 43.38±0.41 | 18.15±1.13 | 23.19±0.54 | 29.83±0.59 | 23.72±0.54 | 15.12±0.69 | 18.78±0.56 | 24.51±0.46 | 19.47±0.37 | 28.86±0.33 |
| | BWT | – | – | – | – | – | – | – | – | – | – | – | – | – |
| HAL | ACC | 45.83±3.56 | 50.19±1.68 | 55.48±1.91 | 50.50±0.88 | 20.74±0.99 | 24.04±1.65 | 26.65±2.98 | 23.81±1.01 | 15.91±1.29 | 17.60±1.74 | 18.84±1.29 | 17.45±1.12 | 30.59±0.58 |
| | BWT | -19.52±2.57 | -15.96±2.43 | -10.81±1.61 | -15.43±0.97 | -20.69±1.36 | -17.55±1.34 | -15.60±1.93 | -17.95±1.15 | -20.63±1.26 | -19.17±1.33 | -17.09±1.42 | -18.96±0.62 | -17.45±0.36 |
| MetaSP | ACC | 72.55±0.82 | 77.29±0.99 | 80.68±0.60 | 76.84±0.71 | 53.68±0.44 | 58.63±0.44 | 63.19±0.72 | 58.50±0.23 | 44.51±0.73 | 50.89±0.65 | 56.98±0.67 | 50.79±0.56 | 62.05±0.33 |
| | BWT | -17.24±0.64 | -11.79±1.48 | -7.83±0.81 | -12.29±0.90 | -23.32±1.18 | -17.20±1.12 | -11.29±1.30 | -17.27±1.06 | -31.76±0.71 | -24.72±0.63 | -17.96±0.67 | -24.81±0.52 | -18.12±0.60 |
| SOIF | ACC | 70.74±1.56 | 71.82±1.38 | 62.16±1.54 | 68.24±1.47 | 51.02±0.66 | 50.78±1.07 | 46.36±0.87 | 49.39±0.57 | 38.10±1.10 | 37.43±1.33 | 31.17±0.72 | 35.57±0.52 | 51.07±0.77 |
| | BWT | -16.36±1.68 | -16.00±1.03 | -27.18±1.54 | -19.85±1.26 | -26.24±0.78 | -26.05±1.48 | -32.03±1.58 | -28.11±0.81 | -37.98±1.48 | -38.56±1.61 | -45.79±0.86 | -40.78±0.55 | -29.58±0.75 |
| CSReL-ER | ACC | 70.11±1.14 | 75.74±0.53 | 80.03±0.50 | 75.29±0.47 | 53.17±0.31 | 57.50±0.70 | 62.23±0.43 | 57.63±0.32 | 45.55±0.82 | 51.30±0.56 | 56.03±0.82 | 50.96±0.43 | 61.30±0.17 |
| | BWT | -19.19±1.09 | -13.06±0.44 | -8.63±0.76 | -13.63±0.49 | -20.56±0.51 | -14.87±0.59 | -10.38±0.38 | -15.27±0.34 | -28.57±1.20 | -22.31±0.82 | -16.94±0.70 | -22.61±0.51 | -17.17±0.20 |
| CSReL-CL-Prv | ACC | 76.61±0.40 | 78.10±1.17 | 78.34±2.06 | 77.68±0.69 | 59.66±0.92* | 62.03±1.60 | 65.28±0.59 | 62.32±0.47* | 53.76±0.79 | 58.31±0.77 | 62.44±0.81 | 58.17±0.16* | 66.06±0.31 |
| | BWT | -10.15±1.71 | -8.69±0.45 | -8.10±2.25 | -8.98±1.13 | -13.25±1.38* | -9.31±2.23 | -5.52±0.82* | -9.36±0.69* | -19.19±1.02* | -14.56±0.98 | -9.76±0.56* | -14.50±0.70* | -10.95±0.66 |
| CGR | ACC | 77.50±0.56* | 80.24±0.44* | 83.19±0.53* | 80.31±0.18* | 58.18±0.41 | 61.80±0.37 | 64.82±0.35 | 61.60±0.35 | 53.14±0.49 | 57.92±0.30 | 61.91±0.45 | 57.66±0.24 | 66.52±0.10 |
| | BWT | -8.83±1.63 | -5.93±2.18 | -2.49±2.07* | -5.75±1.90* | -15.39±0.51 | -11.07±0.67 | -7.67±0.96 | -11.38±0.61 | -20.59±0.60 | -15.86±0.78 | -11.35±0.88 | -15.93±0.70 | -11.02±0.39 |

in Table 7, CGR runs at roughly 2–4× lower computational cost than CSReL-CL-Prv across datasets, which is important in resource-constrained scenarios.

**Modularity of CGR.** To demonstrate the compatibility and modularity of CGR, we integrate it into two representative rehearsal-based methods: PCR (Lin et al., 2023), and ER-LAS (Huang et al., 2023), both of which employ reservoir sampling. As shown in Table 4, ~~incorporating CGR consistently improves performance across all scenarios~~ **incorporating CGR improves mean performance in nearly all scenarios** in both CIL and TIL settings. The only exception occurs for ER-LAS under the Split CIFAR-100 dataset in the CIL setting with a buffer size of 2000, where the ACC remains nearly identical to the baseline.

Table 4: Performance gains from integrating CGR into PCR and ER-LAS.

| Method | Dataset | Split CIFAR-100 | | | | Split Mini-ImageNet | | | |
|---|---|---|---|---|---|---|---|---|---|
| | Setting | CIL | | TIL | | CIL | | TIL | |
| | Buffer | 1000 | 2000 | 1000 | 2000 | 1000 | 2000 | 1000 | 2000 |
| PCR | ACC | 33.03 | 37.63 | 68.67 | 75.90 | 20.20 | 24.77 | 48.82 | 53.48 |
| | BWT | -48.70 | -49.98 | -19.59 | -13.10 | -57.65 | -52.41 | -22.31 | -17.29 |
| PCR-CGR | ACC | **36.83** | **42.24** | **72.48** | **76.94** | **29.22** | **32.25** | **52.87** | **57.49** |
| | BWT | **-31.46** | **-24.69** | **-15.31** | **-10.57** | **-41.26** | **-36.96** | **-18.29** | **-12.58** |
| ER-LAS | ACC | 37.61 | **45.69** | 78.84 | 82.30 | 28.80 | 34.73 | 56.64 | 60.52 |
| | BWT | -49.48 | -37.88 | -10.02 | -6.58 | -46.01 | -37.82 | -15.09 | -11.93 |
| ER-LAS-CGR | ACC | **39.85** | 45.62 | **80.26** | **82.41** | **34.12** | **39.34** | **61.85** | **65.13** |
| | BWT | **-43.72** | **-33.43** | **-7.46** | **-4.86** | **-40.30** | **-30.30** | **-11.00** | **-7.25** |

**Compatibility with Other CL Pipelines.** CGR is intended as a buffer-update policy rather than a complete CL objective. Therefore, it can be combined with rehearsal-based CL methods that maintain an explicit sample buffer and train a classifier with labeled current-task data. In such methods, CGR can replace or augment the default memory update rule by ranking current-task examples according to target-confidence variance, while leaving the method's loss, replay retrieval strategy, and optimization procedure unchanged. This is why CGR can be integrated with ER-based methods such as PCR and ER-LAS, as shown in our experiments. However, CGR is not directly applicable without modification to methods that do not store raw examples, methods based purely on generative replay, task-free single-pass streaming

methods, or unsupervised/reinforcement-learning settings where target-class confidence is unavailable. **Extending the confidence-variance proxy to these settings would require alternative uncertainty or consistency signals, which we leave for future work.**

**Confidence-Based Update Policies.** In our method, by epoch $E$, we measure the variance of target confidence for each sample to identify the most challenging ones within each class, while maintaining balanced class and task distributions in the buffer. Table 5 compares CGR with alternative update strategies: selecting samples with the highest (simplest) and lowest (hardest) mean target confidence across epochs $E$, as well as uniform sampling—each under class and task balance constraints. As shown, ~~CGR outperforms these alternatives~~ **CGR achieves higher mean ACC and less negative BWT than these alternatives in nearly all scenarios** in both CIL and TIL settings.

Table 5: Comparing CGR with three update policies—highest mean confidence (simplest samples), lowest mean confidence (hardest samples), and uniform sampling—under class- and task-balanced conditions in both CIL and TIL settings.

| Method | Dataset | Split CIFAR-100 | | | | Split Mini-ImageNet | | | | Mean | |
|---|---|---|---|---|---|---|---|---|---|---|---|
| | Setting | CIL | | TIL | | CIL | | TIL | | CIL | TIL |
| | Buffer | 1000 | 2000 | 1000 | 2000 | 1000 | 2000 | 1000 | 2000 | | |
| Uniform | ACC | 32.62 | 40.79 | 79.11 | 82.36 | 27.53 | 33.90 | 58.64 | 62.24 | 33.71 | 70.59 |
| | BWT | -52.74 | -41.53 | -6.74 | -3.17 | -50.87 | -39.51 | -15.32 | -10.15 | -46.16 | -8.85 |
| Hardest | ACC | 19.72 | 25.86 | 65.36 | 70.33 | 18.56 | 22.13 | 43.44 | 48.72 | 21.57 | 56.96 |
| | BWT | -67.90 | -59.02 | -22.21 | -16.66 | -62.30 | -55.27 | -33.95 | -27.01 | -61.12 | -24.96 |
| Simplest | ACC | 32.59 | 38.99 | 80.01 | 82.17 | 29.35 | 35.32 | 60.81 | 64.17 | 34.06 | 71.79 |
| | BWT | -53.30 | -44.33 | -5.78 | -3.24 | -48.73 | -38.74 | -12.44 | -8.33 | -46.28 | -7.45 |
| CGR | ACC | **34.65** | **41.03** | **80.24** | **83.19** | **30.05** | **36.36** | **61.80** | **64.82** | **35.52** | **72.51** |
| | BWT | **-51.34** | **-41.53** | -5.93 | **-2.49** | **-47.38** | **-37.17** | **-11.07** | **-7.67** | **-44.36** | **-6.79** |

**Running Time.** In Tables 6 and 7, we report per-hour running times of various methods using a Quadro RTX 8000 GPU. Table 6 compares CGR with other methods on Split CIFAR-100 (buffer size: 2000). As shown, CGR is as fast as ER and outperforms others in runtime efficiency.

Table 6: Running times (hours) for methods on Split CIFAR-100 (buffer size: 2000), using a Quadro RTX 8000 GPU.

| Method | A-GEM | ER | GSS | HAL | MetaSP | SOIF | CSReL-ER | CSReL-CL-Prv | CGR |
|---|---|---|---|---|---|---|---|---|---|
| Runtime | 1.85h | 1.4h | 15.56h | 2.44h | 2.87h | 8.13h | 2.08h | 2.52h | **1.4h** |

To further analyze the computational cost of CSReL-CL-Prv, Table 7 extends the comparison to Split Mini-ImageNet and Split Tiny-ImageNet, using ER as the baseline.

As all datasets were resized to $32 \times 32$ pixels in our experiments, Split Tiny-ImageNet is approximately twice the size of both Split CIFAR-100 and Split Mini-ImageNet. The results show that CGR and ER maintain consistent runtimes when moving from Split CIFAR-100 to Split Mini-ImageNet. In contrast, CSReL-CL-Prv's cost increases significantly. On Split Tiny-ImageNet, both CGR and ER roughly double in runtime, while CSReL-CL-Prv increases by several times.

CGR matches or surpasses ER in runtime efficiency because it updates the buffer once per task—after completing each task's training—rather than batch-by-batch as in reservoir sampling. As a result, the buffer is empty during the first task, making CGR even faster than ER during early training.

**Why CGR is Faster.** **The runtime gap between CGR and CSReL-CL-Prv has a simple structural cause. CGR's buffer update consists of one forward pass on each current-task batch during the first $E \in [2, 5]$ epochs to record target-class confidences (line 9 of Algorithm 1), followed by a single sort over per-class confidence variances at the end of each task—no auxiliary models, no backward passes, and no per-task repeated selection. CSReL-CL-Prv, by contrast, performs three additional operations per task transition (Tong et al. (2025), Algorithms 3 and**

Table 7: Running times (hours) for ER (used as the baseline), CSReL-CL-Prv, and CGR across all three datasets (buffer size: 2000), evaluated on a Quadro RTX 8000 GPU.

| Dataset | Method | ER | CSReL-CL-Prv | CGR |
|---|---|---|---|---|
| **Split CIFAR-100** | | 1.4h | 2.52h | **1.4h** |
| **Split Mini-ImageNet** | **Runtime** | 1.4h | 5.37h | **1.4h** |
| **Split Tiny-ImageNet** | | 2.8h | 11.03h | **2.7h** |

**4): (i) it trains a separate *holdout model* on the current task plus a memory subsample (Tong et al. (2025), Algorithm 3, holdout step); (ii) it then runs a greedy selection procedure that performs $t_{\mathbf{out}}$ rounds of subset training and candidate scoring (Tong et al. (2025), Algorithm 3, loop); and (iii) crucially, it repeats this greedy selection *once per previously seen task* to re-shrink each task's coreset to the new per-task budget (Tong et al. (2025), Algorithm 4, inner loop). The extra holdout training, the greedy loop, and the per-previous-task repetition together cause CSReL-CL-Prv's overhead to grow much faster than CGR's, especially as $t_{\mathbf{out}}$ increases. This explains the pattern in Table 7.**

**Bookkeeping Cost of Confidence Trajectories.** A natural concern is whether recording per-sample target confidences across the first $E$ epochs introduces non-trivial memory or time overhead, particularly for larger models or datasets. We quantify this cost in three components. *Memory:* A direct implementation stores one float32 scalar per current-task sample per recorded epoch, totaling $E \cdot N^t \cdot 4$ bytes for a task with $N^t$ examples. For Split Tiny-ImageNet ($N^t \approx 10^4$ per task, $E = 3$), this is approximately $120$ KB, about $0.3\%$ of a ResNet18 backbone ($\approx 45$ MB). The recorded statistic is one scalar per sample regardless of architecture, so this bookkeeping does not scale with model size. Moreover, the full trajectory need not be stored: the variance can be computed online by maintaining only $\sum_e \Gamma_e(x_i, y_i)$ and $\sum_e \Gamma_e(x_i, y_i)^2$ for each sample, reducing memory to $\mathcal{O}(N^t)$ scalars independently of $E$. *Time per training step:* In our implementation, line 9 of Algorithm 1 performs one additional forward pass on the current-task batch to record target-class confidences during the first $E$ epochs; alternatively, the same statistic can be obtained by indexing the current-task entries of the logits already produced by the SGD step in line 11, in which case the cost reduces to a scalar read per sample. Since the SGD update operates on the concatenation of the current-task and buffer batches, the extra forward pass in our implementation applies only to the current-task half-batch and introduces no additional backward pass. With $E = 3$–5 and $m = 50$, this affects $E/m \approx 6\%$–$10\%$ of training epochs, corresponding to an upper bound of roughly $3\%$–$5\%$ of total forward-pass computation under equal current/buffer batch sizes, and a smaller fraction of total training time. *End-of-task selection:* At the end of each task, CGR computes one scalar variance per sample and performs per-class top-$k_{\mathbf{add}}$ and bottom-$k_{\mathbf{prune}}$ selection via a single sort, costing at most $\mathcal{O}(N^t \log N^t)$ scalar operations and completing in milliseconds at the dataset sizes considered here. Together, these components show that CGR has a small bookkeeping overhead, independent of the number of model parameters and not requiring auxiliary models, gradient-based selection, influence estimation, or additional backward passes.**

### 5.3 Ablation

**Why Early-Epoch Confidence Variance Identifies Boundary-Critical Samples.** CGR's central claim has two components: (a) high target-confidence variance during early training selects samples that lie near class decision boundaries, and (b) this variance signal reflects data structure rather than stochastic optimization noise. We support these claims with theoretical and empirical arguments.**

***(a.1) A theoretical link from variance to intermediate confidence.* The per-sample target confidence $\Gamma_e(x_i, y_i) \in [0, 1]$ is bounded, so by the standard variance bound for $[0, 1]$-bounded**

random variables,

$$\sigma^2(x_i, y_i) \leq \bar{\Gamma}(x_i, y_i)\left(1 - \bar{\Gamma}(x_i, y_i)\right),$$

with the right-hand side maximized at $\bar{\Gamma} = 1/2$ and equal to zero at $\bar{\Gamma} \in \{0, 1\}$. A sample can therefore have high variance only if its mean target confidence is bounded away from both extremes—i.e., its predictions fluctuate around an intermediate value, the softmax-confidence analog of low margin in a classifier. This separation is verified empirically in Figure 3: sub-figure (b.1) shows that high-variance samples cluster at intermediate $\bar{\Gamma} \approx 0.5$ while low-variance samples concentrate at $\bar{\Gamma} \to 1$ or $\bar{\Gamma} \to 0$, and sub-figure (a) shows that the high-variance samples are also visually ambiguous across CIFAR-10 classes.

*(a.2) Connection to margin theory.* The role of training-time margin in generalization is well-established in classical margin-based bounds (Bartlett & Mendelson, 2002): classifiers with larger margins on training data admit tighter generalization guarantees. The bound in (a.1) provides a corresponding per-sample statement: high target-confidence variance forces the model's predictions toward intermediate target confidence, the per-sample analog of low margin.

*(b.1) Cross-seed reproducibility.* To test whether CGR's variance signal is dominated by stochastic optimization noise, we measured the rank correlation of per-sample variance vectors across the five seeds used for our main results, on Split CIFAR-100 with buffer 1000. The mean Spearman correlation between seed pairs was $\bar{\rho} = 0.49 \pm 0.01$ over the ten seed pairs—well above the chance-level value of $0$, and remarkably consistent across pairs. This indicates that the variance ranking carries substantial data-dependent structure rather than being dominated by stochastic optimization noise.

*(b.2) Diagnostic comparison of selection rules.* To test whether CGR's selection identifies boundary-sensitive rather than merely unstable samples, we compare the samples selected by CGR's variance criterion against four alternatives—random, high-loss, high-confidence, and low-confidence—on Split CIFAR-100 (buffer 1000) after the first task, averaged over 5 seeds. For each rule, on the selected samples, we report in the following Table the mean target margin $p_{y_{\text{true}}}(x) - \max_{y' \neq y_{\text{true}}} p_{y'}(x)$ and mean target confidence over the first $E$ epochs (matching CGR's selection window), and the mean number of forgetting events (Toneva et al., 2018) over all task-1 epochs. The five rules separate cleanly into three regimes: a *failure regime* (Low confidence and High loss, with strongly negative mean margin of $\approx -0.36$ and high forgetting), an *easy regime* (High confidence, with the highest margin of $+0.37$ and lowest forgetting), and Random, which sits near zero on margin. CGR's selection occupies a position that no other rule produces: positive mean margin ($+0.184$) but lower than the easy regime, intermediate mean confidence ($0.443$, between Random's $0.302$ and High confidence's $0.562$), and forgetting events ($3.07$) lower than Random ($3.99$). This shows that CGR-selected samples are (i) on average correctly classified by the model (positive margin), distinguishing them from persistent failures; (ii) uncertain (intermediate confidence), distinguishing them from easy samples; and (iii) more stable across training than Random, distinguishing them from arbitrary instability. The combination is consistent with informative, learnable boundary samples.

Comparison of different sample-selection rules using margin, forgetting events, and target confidence statistics.

| Selection rule | Mean margin | Forgetting events | Mean target confidence |
|---|---|---|---|
| Random | $-0.017 \pm 0.009$ | $3.99 \pm 0.20$ | $0.302 \pm 0.006$ |
| High loss | $-0.354 \pm 0.010$ | $4.93 \pm 0.22$ | $0.097 \pm 0.009$ |
| High confidence | $+0.368 \pm 0.019$ | $2.54 \pm 0.27$ | $0.562 \pm 0.013$ |
| Low confidence | $-0.367 \pm 0.008$ | $4.97 \pm 0.20$ | $0.072 \pm 0.003$ |
| High variance (CGR) | $+0.184 \pm 0.017$ | $3.07 \pm 0.24$ | $0.443 \pm 0.008$ |

*(b.3) Precedent for early-training importance scores.* Paul et al. (2021) introduced gradient- and error-norm scores computable at very early training epochs and demonstrated that these scores reliably identify samples whose removal harms generalization. Their results provide

precedent for the broader principle on which CGR rests: early-training importance scores can carry meaningful, reproducible information about training samples.

Taken together, (a.1) and (a.2) establish that high confidence variance corresponds to intermediate-confidence, low-margin behavior consistent with boundary sensitivity, and (b.1)–(b.3) provide direct empirical evidence that the signal is reproducible across runs and selects qualitatively different samples than each of the four alternative selection rules considered. We also acknowledge that other sources of instability—such as label noise—may induce high variance; we evaluate this case in the next paragraph.

**Robustness to Label and Target Noise.** A natural concern is that variance-based selection may be sensitive to target noise: samples whose predicted-target variance is high not because they are near a decision boundary, but because they are intrinsically noisy, mislabeled, or stochastically ambiguous (an instance of the "noisy TV" failure mode in curiosity-driven learning, where high prediction error reflects irreducible randomness rather than learnable signal (Pathak et al., 2017; Burda et al., 2018)). CGR mitigates this in two ways. *(i) Early-epoch window.* Variance is computed only over the first $E = 2$–$5$ epochs of each task, before the network has begun memorizing noisy labels—a phenomenon shown to occur predominantly in late training (Zhang et al., 2016; Arpit et al., 2017). Mislabeled samples therefore tend to exhibit consistently low target confidence in this early window, falling into the low-variance "hard outlier" region (Area 3 in Figure 3, sub-figure b.1) which CGR explicitly excludes; clean boundary samples in contrast show transient uncertainty that resolves over training (Area 2 collapses into Area 4 in Figure 3, sub-figure b.2). It is precisely this resolvable uncertainty signature that CGR captures, in contrast to the irreducible uncertainty of noisy targets. *(ii) Per-class and per-task buffer balance.* CGR enforces a fixed budget of $B_{\text{size}}/(t \cdot |\mathcal{C}^t|)$ slots per class, bounding the fraction of the buffer that any single class—including a noisy one—can occupy. We expect the early-epoch and class-balance design to provide partial robustness, but caution that under sufficiently high noise, the variance signal could degrade, and pairing CGR with an explicit noise-detection mechanism (e.g., loss-based filtering as in Arazo et al. (2019)) is a natural extension.

To complement the discussion above, we evaluate CGR under symmetric within-task label noise on Split CIFAR-100 (buffer size 1000), with noise rates $\eta \in \{0, 0.1, 0.2\}$ using ACC averaged over five seeds. We compare against three baselines: ER, CSReL-CL-Prv (reducible-loss selection), and the Uniform baseline that draws samples uniformly at random under the same class- and task-balance constraints as CGR. The Uniform baseline isolates the selection criterion of CGR (variance signal) from the effect of balancing. Three observations follow from the following Table. First, CGR's advantage over ER on CIL is essentially flat across noise levels ($+10.2$, $+9.3$, $+8.4$ at $\eta = 0, 0.1, 0.2$), indicating that the variance signal does not collapse at the noise levels tested. Second, against the Uniform baseline — which holds balance fixed and changes only the selection rule — CGR's gap increases with noise on both CIL ($+2.0 \rightarrow +3.7 \rightarrow +4.5$) and TIL ($+1.1 \rightarrow +5.9 \rightarrow +9.8$). The variance criterion thus contributes meaningfully beyond what balance alone provides, and this contribution does not erode in the tested regime. Third, CGR remains the best method on TIL at every noise level and best on CIL at $\eta = 0$. At $\eta = 0.2$, however, CSReL-CL-Prv overtakes CGR on CIL (22.70 vs. 20.11). This is consistent with the discussion above that explicit loss-based filtering becomes increasingly useful as label noise grows, and matches the natural extension to CGR outlined earlier.

Comparison under symmetric within-task label noise on Split CIFAR-100 (buffer 1000), evaluated using ACC.

| Method | $\eta = 0$ | | $\eta = 0.1$ | | $\eta = 0.2$ | |
|---|---|---|---|---|---|---|
| | CIL | TIL | CIL | TIL | CIL | TIL |
| ER | 24.46 | 78.12 | 15.87 | 61.37 | 11.76 | 48.72 |
| CSReL-CL-Prv | 32.27 | 78.10 | **26.00** | 69.03 | **22.70** | 61.87 |
| Uniform | 32.62 | 79.11 | 21.47 | 63.47 | 15.58 | 52.20 |
| CGR | **34.65** | **80.24** | 25.12 | **69.34** | 20.11 | **62.02** |

**Class Imbalance.** CGR uses class-balanced memory allocation in the reported experiments because the standard Split CIFAR-100, Split Mini-ImageNet, and Split Tiny-ImageNet benchmarks have balanced task/class structure. This design ensures that no previously seen class is removed from memory solely because of stream order or class frequency. However, this allocation is not the only possible choice. In imbalanced streams, CGR can be combined with alternative allocation rules, such as proportional allocation according to observed class frequency, equal allocation for balanced accuracy, or hybrid allocation with a minimum quota per class; we leave a systematic evaluation of such variants to future work. The confidence variance score itself is independent of the allocation rule; the allocation rule determines how many samples are retained from each class, while CGR ranks samples within the allocated class-specific budget.

**Results on Split Mini-ImageNet at the Original Resolution ($84 \times 84$).** Table 8 reports results on Split Mini-ImageNet for both CIL and TIL settings, using the original image resolution of $84 \times 84$ pixels (as opposed to the downsampled $32 \times 32$) and a buffer size of 1000. This evaluation is designed to assess the impact of image downscaling on model performance. As shown, our method still achieves comparable results to the baselines in terms of both ACC and BWT across CIL and TIL settings. Furthermore, as expected, increasing the resolution to $84 \times 84$ leads to ACC improvements for all methods.

Table 8: Results on Split Mini-ImageNet at the original $84 \times 84$ resolution with a buffer size of 1000. ACC and BWT are reported for both CIL and TIL settings. Higher-resolution inputs improve ACC for all methods, and our approach remains competitive across metrics.

| Dataset: Split Mini-ImageNet [$84 \times 84$] — buffer size: 1000 | | | | | | | | | | |
|---|---|---|---|---|---|---|---|---|---|---|
| Setting | Method | A-GEM | ER | GSS | GDUMB | HAL | MetaSP | SOIF | CSReL-ER | CSReL-CL-Prv | CGR |
| CIL | *ACC* | 16.48 | 26.57 | 24.34 | 13.71 | 8.65 | 35.09 | 29.14 | 29.19 | 35.09 | 35.41 |
| | *BWT* | -76.84 | -65.52 | -66.45 | - | -45.48 | -54.47 | -63.29 | -63.34 | -51.21 | -51.28 |
| TIL | *ACC* | 40.65 | 66.00 | 60.28 | 29.07 | 27.75 | 67.51 | 65.06 | 67.95 | 71.03 | 70.40 |
| | *BWT* | -46.63 | -16.58 | -23.81 | - | -22.55 | -16.38 | -19.48 | -15.26 | -8.40 | -9.44 |

**Final Class and Task Distribution in the Buffer.** Table 9 presents an analysis of the final buffer composition for each method, highlighting potential long-tailed memory effects (Samuel & Chechik, 2021; Shi et al., 2023). At the end of training, we record the number of stored samples per class and per task, and compute the Coefficient of Variation (CV) as $\text{CV} = \left(\frac{\text{Standard Deviation}}{\text{Mean}}\right) \times 100\%$. A CV of 0.00 indicates a perfectly balanced distribution, while higher values reflect increasing imbalance and long-tailed memory bias. Our CGR method achieves a CV of 0.00 for both class and task distributions, demonstrating perfectly balanced memory allocation. Similarly, A-GEM and CSReL-CL-Prv also exhibit task-level CVs of 0.00, indicating ideal task balance. ER, HAL, and CSReL-ER yield CVs close to zero, suggesting near-uniform task distributions. In contrast, GSS, MetaSP, and SOIF display substantially higher CV values, reflecting pronounced task-level imbalance. Regarding class distribution, only our method achieves a CV of 0.00, confirming perfect inter-class balance, whereas all other methods show higher CVs, indicating residual class imbalance within the replay buffer.

Table 9: Coefficient of Variation (CV) for class and task distributions in the buffer after training across different methods, using a buffer size of 1000 with Split CIFAR-100.

| Method | A-GEM | ER | GSS | HAL | MetaSP | SOIF | CSReL-ER | CSReL-CL-Prv | CGR |
|---|---|---|---|---|---|---|---|---|---|
| CV of Tasks | 0.00 | 06.80 | 58.79 | 06.21 | 26.96 | 29.87 | 04.56 | 0.00 | **0.00** |
| CV of Classes | 29.77 | 31.43 | 68.38 | 34.35 | 53.85 | 63.97 | 37.47 | 44.25 | **0.00** |

**Hyperparameter Sensitivity.** In Figure 3, sub-figures (c.1) and (c.2), we examine the sensitivity of our hyperparameter $E$, which is integral to our memory update policy. As illustrated, the performance of hyperparameter $E$ remains stable across a range from 2 to 6 on all three datasets, maintaining consistency in

both ACC and BWT. Additionally, the results indicate that our approach for identifying boundary samples requires only a few initial training epochs.

# 6 Conclusion

While rehearsal-based methods have shown strong performance in Continual Learning (CL), their effectiveness is often limited by computational constraints and reliance on expensive buffer update strategies. Our proposed method, Confidence-Guided Replay (CGR), addresses these limitations by introducing a lightweight, forward-pass-only approach that tracks target confidence variability to identify boundary-critical samples. By leveraging the temporal dynamics of sample confidence within the main model, CGR enhances decision boundary robustness and mitigates forgetting— ~~without incurring~~ **at minimal** additional computational cost. This principled and efficient strategy offers a practical alternative to existing rehearsal policies, paving the way for more scalable and adaptive CL systems. **CGR currently targets supervised classification scenarios with clear task boundaries and multi-epoch access to each task's data. It is not directly designed for strict online, task-free, regression, unsupervised, or reinforcement-learning-based CL settings. Extending confidence-variance-based replay to task-free or single-pass online CL would require replacing task-level buffer updates with streaming confidence statistics, which we leave for future work.**

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

# A  Appendix

## A.1  Implementation Details

For the CSReL-ER hyperparameters, on Split CIFAR-100, we set the main continual learner's learning rate to 2e-2, the cross-entropy (CE) loss factor to 1.0, and the selection steps to 100. For the selected subset of the current task, the learning rate and number of training steps were set to 5e-3 and 20, respectively. On Split Mini-ImageNet and Split Tiny-ImageNet, we used a main continual learner learning rate of 3e-2 and kept the CE loss factor at 1.0. The selection steps were set to 250. The learning rate and training steps for the selected subset in these datasets were also set to 5e-3 and 20, respectively.

For the CSReL-CL-Prv hyperparameters, on Split CIFAR-100, we set the main continual learner's learning rate to 2e-2, the loss factor to 4.0, and the selection steps to 40. For the selected subset, the learning rate and number of training steps were set to 5e-3 and 7, respectively. For the holdout model, we set the number of epochs, learning rate, and samples per task from previous tasks to 10, 3e-3, and 200, respectively. On

Split Mini-ImageNet and Split Tiny-ImageNet, we used a main continual learner learning rate of 2e-2 and kept the loss factor at 4.0. The selection steps were set to 250. The learning rate and training steps for the selected subset in these datasets were set to 5e-3 and 20, respectively. For the holdout model, we set the number of epochs, learning rate, and samples per task from previous tasks to 20, 3e-3, and 200, respectively.

**For the baselines, we use the values reported in their respective original papers and the Mammoth framework, without additional tuning. The only method-specific hyperparameter introduced by CGR is $E$, the number of early epochs over which the per-sample target-confidence variance is computed. We searched $E$ over the range $\{2, 3, 4, 5, 6, 7, 8, 9, 10\}$ for each (dataset, buffer-size) configuration. The selected values are summarized in Table 10.**

Table 10: Selected values of the CGR hyperparameter $E$ per dataset and buffer size.

| Dataset | Buffer 500 | Buffer 1000 | Buffer 2000 |
|---|---|---|---|
| Split CIFAR-100 | 3 | 4 | 5 |
| Split Mini-ImageNet | 3 | 3 | 4 |
| Split Tiny-ImageNet | 3 | 3 | 3 |

## A.2 Statistical Significance of CGR vs. CSReL-CL-Prv

**Tables 11 and 12 report paired two-sided $t$-tests ($\alpha = 0.05$, $n = 5$ matched seeds) comparing CGR against the strongest competitor, CSReL-CL-Prv, for every (dataset, buffer) cell of Tables 2 and 3, as well as for per-dataset and overall pooled comparisons. Pairing is performed on (dataset, buffer, seed) so that each test contrasts two runs that differ only in method. The * markers in Tables 2 and 3 correspond directly to the entries in these tables. In summary, CGR's ACC advantage in CIL is significant in 12 of 13 cells; its CIL-BWT advantage is significant in 3 of 13 individual cells but reaches significance in the overall pooled comparison ($p = 0.010$); in TIL, CGR is significantly stronger on Split CIFAR-100 (ACC and BWT), while CSReL-CL-Prv is significantly stronger on Split Mini- and Tiny-ImageNet, and the overall pooled comparison shows no significant difference for either metric.**

Table 11: CIL; Paired two-sided t-test: CGR vs CSReL-CL-Prv; $H0$: mean(CGR - CSReL) = 0; $H1$: mean(CGR - CSReL) $\neq$ 0; $\alpha = 0.05$.

| Metric | Dataset | Split CIFAR-100 | | | | Split Mini-ImageNet | | | | Split Tiny-ImageNet | | | | ALL |
|---|---|---|---|---|---|---|---|---|---|---|---|---|---|---|
| | Buffer | 500 | 1000 | 2000 | All | 500 | 1000 | 2000 | All | 500 | 1000 | 2000 | All | |
| $t$-statistic | ACC | 4.6826 | 1.3970 | 3.3156 | 3.6979 | 2.8864 | 2.9459 | 3.1725 | 5.0278 | 5.0929 | 10.1104 | 9.1926 | 8.3536 | 6.3545 |
| | BWT | 0.8965 | 0.0201 | 1.2556 | 1.3604 | 1.6394 | 1.0403 | -0.7704 | 1.2414 | 1.2400 | 3.0589 | 2.1066 | 3.6428 | 2.6906 |
| $p$-value | ACC | 0.009429 | 0.234921 | 0.029501 | 0.002387 | 0.044724 | 0.042143 | 0.033778 | 0.000185 | 0.007017 | 0.000539 | 0.000778 | 0.000001 | 0.000000 |
| | BWT | 0.420668 | 0.984932 | 0.277596 | 0.195216 | 0.176464 | 0.356949 | 0.484048 | 0.234857 | 0.282762 | 0.037700 | 0.102888 | 0.002662 | 0.010043 |
| reject $H0$ at $\alpha = 0.05$ | ACC | Yes | No | Yes | Yes | Yes | Yes | Yes | Yes | Yes | Yes | Yes | Yes | Yes |
| | BWT | No | No | No | No | No | No | No | No | No | Yes | No | Yes | Yes |

Table 12: TIL; Paired two-sided t-test: CGR vs CSReL-CL-Prv; $H0$: mean(CGR - CSReL) = 0; $H1$: mean(CGR - CSReL) $\neq$ 0; $\alpha = 0.05$.

| Metric | Dataset | Split CIFAR-100 | | | | Split Mini-ImageNet | | | | Split Tiny-ImageNet | | | | ALL |
|---|---|---|---|---|---|---|---|---|---|---|---|---|---|---|
| | Buffer | 500 | 1000 | 2000 | All | 500 | 1000 | 2000 | All | 500 | 1000 | 2000 | All | |
| $t$-statistic | ACC | 3.0651 | 5.0278 | 6.6481 | 5.0404 | -4.7617 | -0.2739 | -1.1235 | -2.1572 | -1.9680 | -0.9061 | -1.4946 | -2.5757 | 1.4757 |
| | BWT | 1.1861 | 2.5906 | 4.2779 | 4.1135 | -2.9103 | -1.7319 | -3.4849 | -4.6435 | -3.1049 | -1.7887 | -3.6979 | -4.8150 | -0.1608 |
| $p$-value | ACC | 0.037470 | 0.007345 | 0.002658 | 0.000180 | 0.008894 | 0.797731 | 0.324102 | 0.048855 | 0.120449 | 0.416109 | 0.209336 | 0.021994 | 0.147136 |
| | BWT | 0.301230 | 0.060645 | 0.012869 | 0.001054 | 0.043665 | 0.158331 | 0.025239 | 0.000380 | 0.036048 | 0.148173 | 0.020874 | 0.000275 | 0.872985 |
| reject $H0$ | ACC | Yes | Yes | Yes | Yes | Yes | No | No | Yes | No | No | No | Yes | No |
| at $\alpha = 0.05$ | BWT | No | No | Yes | Yes | Yes | No | Yes | Yes | Yes | No | Yes | Yes | No |