# OpenReview forum: "CGR: Confidence-Guided Replay for Buffer-Based Continual Learning"
_TMLR — Rejected by TMLR_

### Review · Reviewer_hXBa · 2026-03-10

**Summary Of Contributions:**

This paper focuses on buffer management for rehearsal-based continual learning. The key idea is to use the variance of model confidence during the early stages of training as a criterion for retaining boundary-critical samples in the replay buffer. The method is evaluated on Split CIFAR-100, Mini-ImageNet, and Tiny-ImageNet, compared with several rehearsal baselines, and supported by runtime comparisons and ablation studies.

**Audience:**

Yes

**Audience Explanation:**

N/A

**Claims And Evidence:**

Yes

**Claims Explanation:**

- I appreciate that the paper targets a real bottleneck in rehearsal-based CL: better sample selection without paying the heavy cost of gradients, influence functions, or extra models. The proposed signal is easy to understand, easy to implement, and the runtime comparison is genuinely favorable. In particular, matching ER-level runtime while improving over stronger replay baselines is a meaningful practical point.

- The empirical section is reasonably solid for this paper’s scope. The method is tested on three standard benchmarks, under both CIL and TIL, across multiple buffer sizes, and the gains are fairly consistent.

- One concern is soundness at the level of justification. The paper repeatedly connects high confidence variance with boundary-criticality and better forgetting mitigation, but this is supported mostly by intuition, a CIFAR-10 visualization, and downstream results rather than a more formal analysis. I was not fully convinced that confidence variance reliably separates “useful ambiguous samples” from other unstable cases across broader CL regimes.

- I was also a bit puzzled by the assumptions behind the balanced buffer design. The presentation explicitly assumes equal numbers of classes and samples per task, and the update happens after each task, which makes the method most natural for task-boundary-aware settings. That is fine for the reported benchmarks, but it narrows the claimed generality. The paper would be stronger if it were clearer about how CGR behaves in less regular streams, imbalanced tasks, or fully online/task-free settings.

- Finally, while the experiments are good overall, the evaluation still feels somewhat bounded. The results are all on image classification benchmarks with a ResNet18 backbone, and the strongest gains are mostly incremental rather than dramatic. For TMLR, I would have liked either a more rigorous analysis of why the proxy works, or broader evidence that it remains robust under harder settings such as class imbalance, noisy labels, or non-stationary streams without clean task segmentation.

**Requested Changes:**

- The method assumes that high early confidence variance identifies boundary-critical and informative replay samples. Could you provide a stronger theoretical justification or more direct empirical evidence that this signal is not merely reflecting optimization instability or noise?
- The runtime results are promising, but the paper sometimes suggests that CGR adds almost no overhead. Could you clarify the bookkeeping cost of storing confidence trajectories over the first E epochs, especially for larger models or datasets?
- Have you evaluated the method under label noise, class imbalance, or more challenging non-prototypical data distributions?
- Could you comment on whether this confidence-variance proxy can be integrated into more recent continual learning pipelines beyond PCR and ER-LAS?

---

> ### Author Response · Authors · 2026-05-02
>
> **Response on the bookkeeping cost of confidence trajectories.** We thank the reviewer for prompting us to clarify this point — we agree the paper should distinguish "no auxiliary training overhead" from "no overhead at all." Section 5.2 ("Bookkeeping Cost of Confidence Trajectories" paragraph) now quantifies CGR's bookkeeping cost in three components. *Memory*: a direct implementation stores $E \cdot N^t$ float32 values per task ($\approx 120$~KB for Split Tiny-ImageNet at $N^t \approx 10^4$, $E=3$, $\approx 0.3\\%$ of the ResNet18 backbone); we also note that the full trajectory need not be stored, since the variance can be computed online from two running statistics per sample, giving $\mathcal{O}(N^t)$ memory independent of $E$ and of model size. *Per-step time*: in our implementation, recording introduces one forward pass on the current-task half-batch during the first $E$ of $m=50$ epochs and no additional backward pass, contributing an upper bound of roughly $3\\%$--$5\\%$ of total forward-pass computation; alternatively, the recording can be implemented by indexing logits already produced by the training forward pass, in which case no extra computation is incurred. *End-of-task selection*: a single $\mathcal{O}(N^t \log N^t)$ sort completing in milliseconds. We have also softened earlier phrasings, such as "without incurring additional computational cost," to be consistent with this analysis.

---

> ### Author Response · Authors · 2026-05-22
>
> **Response on integration with other CL pipelines.** We thank the reviewer for this question. We agree that the manuscript should better discuss how general the confidence-variance proxy is beyond the two plug-in examples evaluated in the paper. In the revised version, we have added a discussion (Section 5.2, after the "Modularity of CGR" paragraph) clarifying that CGR is a buffer-update policy rather than a full continual learning objective. Therefore, it can be integrated into rehearsal-based supervised classification pipelines that maintain an explicit replay buffer and expose target-class probabilities or logits during training. In such cases, CGR can replace or augment the method’s existing buffer update rule while leaving the training loss, replay retrieval strategy, and optimization procedure unchanged. This includes approaches such as ER-style methods. At the same time, we have clarified the limitations: CGR is not directly applicable without modification to pure generative replay, prototype-only methods without stored examples, task-free single-pass online CL, unsupervised CL, or reinforcement-learning settings where target-class confidence is unavailable.
>
> **Response on the soundness of the variance signal.** We thank the reviewer for prompting this analysis. The revised manuscript (Section 5.3, added paragraph “Why Early-Epoch Confidence Variance Identifies Boundary-Critical Samples”) supports the central claim along five converging lines, which together address both halves of the reviewer's question (the variance signal corresponds to boundary samples, and it is not optimization noise).
>
> *Variance corresponds to boundary samples.* The paragraph includes a new theoretical argument: since the target confidence $\Gamma_e(x_i, y_i)$ is bounded in $[0, 1]$, the standard variance bound gives $\sigma^2 \leq \bar\Gamma\(1 - \bar\Gamma)$, which is maximized at $\bar\Gamma = 1/2$ and zero at $\bar\Gamma \in \\{0, 1\\}$. A sample can therefore have high variance only if its mean target confidence is intermediate—the per-sample analog of low margin. This is verified empirically in Figure 3 and connects directly to classical margin theory (Bartlett & Mendelson, 2002).
>
> *Variance is not optimization noise, and selects boundary samples specifically.* (i) The pairwise Spearman correlation of per-sample variance vectors across 5 seeds is $\bar\rho = 0.49 \pm 0.01$ on Split CIFAR-100 with buffer 1000—well above chance and remarkably consistent across the 10 seed pairs, indicating that the ranking is substantially reproducible across independent training runs. (ii) A diagnostic comparison (see following Table) places CGR's selection at a position no other rule produces: CGR-selected samples have positive mean margin ($+0.184$, vs $\approx\\!-0.36$ for Low confidence / High loss and $-0.02$ for Random), intermediate mean confidence ($0.443$), and fewer forgetting events than Random ($3.07$ vs $3.99$). This shows that CGR-selected samples are correctly classified on average (positive margin), uncertain (intermediate confidence), and more stable over training than Random—consistent with informative, learnable boundary samples rather than persistent failures or arbitrary instability. Paul et al. (2021) provide additional precedent that early-training importance scores reliably identify informative examples.
>
> | Selection rule | Mean margin | Forgetting events | Mean target confidence |
> |---|---:|---:|---:|
> | Random | $-0.017 \pm 0.009$ | $3.99 \pm 0.20$ | $0.302 \pm 0.006$ |
> | High loss | $-0.354 \pm 0.010$ | $4.93 \pm 0.22$ | $0.097 \pm 0.009$ |
> | High confidence | $+0.368 \pm 0.019$ | $2.54 \pm 0.27$ | $0.562 \pm 0.013$ |
> | Low confidence | $-0.367 \pm 0.008$ | $4.97 \pm 0.20$ | $0.072 \pm 0.003$ |
> | High variance (CGR) | $+0.184 \pm 0.017$ | $3.07 \pm 0.24$ | $0.443 \pm 0.008$ |
>
> Peter L Bartlett and Shahar Mendelson. Rademacher and gaussian complexities: Risk bounds and structural results. Journal of machine learning research, 3(Nov):463–482, 2002.
>
> Mansheej Paul, Surya Ganguli, and Gintare Karolina Dziugaite. Deep learning on a data diet: Finding important examples early in training. Advances in neural information processing systems, 34:20596–20607, 2021.

---

> ### Author Response · Authors · 2026-05-22
>
> **Response on label noise.** We thank the reviewer for raising this concern, which motivated a new experiment that we believe strengthens the paper. We have expanded the new paragraph “Robustness to Label and Target Noise” (Section 5.3), which also addresses a similar concern raised by another reviewer, to include the empirical results.
>
>
> On Split CIFAR-100 (buffer 1000), we evaluated CGR under symmetric within-task label noise at $\eta \\in \\{0, 0.1, 0.2\\}$ using ACC averaged over five seeds, against ER, CSReL-CL-Prv, and the Uniform baseline that draws samples uniformly at random under the same class- and task-balance constraints as CGR. The Uniform baseline isolates the selection criterion of CGR (variance signal) from the effect of balancing.
>
> The results (see following Table) show: (i) CGR's CIL gap over ER is preserved under noise ($+10.2$, $+9.3$, $+8.4$); (ii) CGR's gap over the Uniform baseline increases with noise on both CIL ($+2.0 \to +3.7 \to +4.5$) and TIL ($+1.1 \to +5.9 \to +9.8$), indicating that the variance criterion itself — once class/task balance is held fixed — contributes more, not less, as noise rises on this dataset; (iii) CGR remains the best method on TIL at every noise level. However, at $\eta = 0.2$ on CIL, CSReL-CL-Prv's reducible-loss criterion overtakes CGR ($22.70$ vs. $20.11$), indicating that loss-based filtering becomes increasingly useful at high label noise and suggesting a natural extension to CGR (e.g., loss-based filtering as in Arazo et al. (2019)).
>
>
> | Method | CIL ($\eta=0$) | TIL ($\eta=0$) | CIL ($\eta=0.1$) | TIL ($\eta=0.1$) | CIL ($\eta=0.2$) | TIL ($\eta=0.2$) |
> |---|---:|---:|---:|---:|---:|---:|
> | ER | 24.46 | 78.12 | 15.87 | 61.37 | 11.76 | 48.72 |
> | CSReL-CL-Prv | 32.27 | 78.10 | **26.00** | 69.03 | **22.70** | 61.87 |
> | Uniform | 32.62 | 79.11 | 21.47 | 63.47 | 15.58 | 52.20 |
> | CGR | **34.65** | **80.24** | 25.12 | **69.34** | 20.11 | **62.02** |
>
>
> Eric Arazo, Diego Ortego, Paul Albert, Noel O’Connor, and Kevin McGuinness. Unsupervised label noise modeling and loss correction. In International conference on machine learning, pp. 312–321. PMLR, 2019.
>
>
> **Response on class imbalance.** Thank you for raising this concern. We agree that the original manuscript did not clearly separate CGR's confidence-variance scoring rule from the balanced buffer allocation used in our benchmark implementation. We have added the new paragraph “Class Imbalance” in Section 5.3 to address this concern.
>
> CGR uses class-balanced memory allocation in the reported experiments because the standard Split CIFAR-100, Split Mini-ImageNet, and Split Tiny-ImageNet benchmarks have balanced task/class structure. This design ensures that no previously seen class is removed from memory solely because of stream order or class frequency. However, this allocation is not the only possible choice. In imbalanced streams, CGR can be combined with alternative allocation rules, such as proportional allocation according to observed class frequency, equal allocation for balanced accuracy, or hybrid allocation with a minimum quota per class; we leave a systematic evaluation of such variants to future work. The confidence variance score itself is independent of the allocation rule; the allocation rule determines how many samples are retained from each class, while CGR ranks samples within the allocated class-specific budget.
>
>
> **Response on task-free / online streams.** Thank you for pointing out this concern. CGR is designed for supervised classification-based continual learning (CL) with identifiable task boundaries and multi-epoch access to the current task data. This is because the method computes the temporal variance of the model’s target-class confidence over the first $E$ epochs and performs buffer updates at task transitions. Therefore, the current formulation is directly applicable to task-incremental and class-incremental supervised classification settings, but not to strict single-pass online CL, task-free CL, regression, unsupervised CL, or reinforcement-learning settings without modification.
>
> In the revised manuscript, we have added a “Scope and Assumptions” paragraph in Section 4 and softened the abstract, introduction, and conclusion to avoid suggesting that CGR applies to all CL settings. We have stated that CGR is intended for supervised classification-based CL with clear task boundaries, matching the benchmarks evaluated in the paper. We have also discussed broader settings, such as task-free or online CL, as future work rather than as direct applications of the current method.

---

> > ### Comment · Reviewer_hXBa · 2026-05-26
> >
> > I appreciate the authors’ detailed response. I believe most of my concerns have been addressed, and I have no further questions.

---

### Review · Reviewer_V4MJ · 2026-03-17

**Summary Of Contributions:**

**Summary**

The paper studies a buffer-based continual learning framework in which samples from previous tasks are stored and replayed while learning the current task to mitigate catastrophic forgetting. The authors propose retaining samples that exhibit high variance in prediction confidence during training. They argue that the proposed method is computationally more efficient than existing rehearsal-based continual learning approaches. The efficiency and performance of the proposed method are evaluated on continual learning benchmarks using the CIFAR-100 and Mini-ImageNet datasets, with average accuracy and backward transfer as evaluation metrics.


**Strength**

1. The proposed method is simple and clear--assigning samples with confidence based on tracking the variance of the confidence. The method is computationally efficient compared to other rehearsal-based approaches.

2. The authors show interesting observations from the proposed methods (which aligns with known results) that early training results acts as a generalization result.

**Weakness**

1. The paper is primarily experimental. While theoretical results may not be necessary, the paper could be improved by addressing a few points. For example, the authors could provide a brief theoretical explanation of why the running time of the proposed method is lower than that of CSReL.

2. The methodological comparison with other rehearsal-based methods would be clearer if the authors summarized the results in a table.

**Audience:**

Yes

**Audience Explanation:**

Catastrophic forgetting is an important challenge in continual learning, and the authors propose a simple method to address this problem in a way that has not been previously explored.

**Claims And Evidence:**

Yes

**Claims Explanation:**

The authors conduct experiments to support their claims in CIFAR-100 and Mini-ImageNet datasets.

**Requested Changes:**

1. The preliminaries section can be improved :

- Clarification on mathematical aspects : Does the training samples belong to Euclidean space? Is the loss function smooth ( the authors use cross entropy in their pseudo-code)?


2. $\mathcal{P}$ has not been defined in (2).

---

> ### Author Response · Authors · 2026-05-02
>
> **Response on theoretical explanation of the runtime gap.** We thank the reviewer for the suggestion. Section 5.2 (after “Running Time” paragraph) now includes a brief structural explanation of why CGR is faster than CSReL-CL-Prv. CGR's per-task overhead for sample selection is one forward pass per current-task batch during the first $E\in[2,5]$ epochs plus a single sort at the end of the task. CSReL-CL-Prv, in contrast, performs three additional operations per task transition (Tong et al., 2025, Algorithms 3 and 4): (i) per-task training of a separate holdout model on the current task plus a memory subsample; (ii) a greedy selection loop with $t_\text{out}$ rounds of subset training and candidate scoring; and (iii) repetition of this greedy selection for every previously seen task on each task transition, to re-shrink each task's coreset to the new per-task budget. We argue that this structural asymmetry explains the observed pattern in Table 7.
>
> Ruilin Tong, Yuhang Liu, Javen Qinfeng Shi, and Dong Gong. Coreset selection via reducible loss in continual learning. In The Thirteenth International Conference on Learning Representations, 2025.
>
> **Response on the methodological comparison table.** We thank the reviewer for this suggestion, which was very useful in clarifying CGR's positioning relative to prior work. In the revised manuscript we have added Table 1 in Section 2 (Related Work), summarizing all rehearsal-based CL methods used in our experiments along five methodological dimensions of the buffer-update step: (i) the selection signal driving the buffer update, (ii) whether the selection avoids gradient, Hessian, or influence-function computations, (iii) whether selection avoids training a separate auxiliary model, (iv) whether the buffer policy enforces class- and task-level balance, and (v) the qualitative cost of the buffer-update step itself, independent of training-time overhead. We deliberately scope every column to the buffer-update step rather than to total runtime, since CGR's contribution is a buffer-update policy, and conflating selection cost with training-time costs would obscure the comparison. The table makes explicit the methodological niche CGR occupies: it is the only method in the comparison that simultaneously uses an informativeness-aware selection signal (target-confidence variance), avoids gradient and influence-function computations during selection, requires no auxiliary network, enforces both class- and task-balanced memory, and operates at low selection cost.
>
> **Response on mathematical clarification in the Preliminaries.** We thank the reviewer for the suggestion. Section 3 (Preliminaries) has been revised to make the mathematical setup explicit: (i) the input space is now stated as $\mathcal{X} \subseteq \mathbb{R}^d$, with $d$ given for the benchmarks used, and the model is written as $f_\theta : \mathcal{X} \to \mathbb{R}^{|\mathcal{C}|}$ outputting class logits; (ii) labels $y_i^t \in \mathcal{C}^t$ and the disjoint-class condition $\mathcal{C}^t \cap \mathcal{C}^{t'} = \emptyset$ are now stated; (iii) the loss $\ell$ is now explicitly identified as the standard softmax cross-entropy, characterized as smooth and bounded below by zero as a function of the logits, which justifies SGD-based optimization. These edits also align Section 3 with the cross-entropy used in Algorithm 1.
>
> **Response on undefined $\mathcal{P}$ in Eq. 2.** We thank the reviewer for catching this. We have rewritten Eq. 2 to use explicit softmax notation, $\Gamma(x_i^t, y_i^t) = \Big[\mathrm{softmax}\left(f_\theta(x_i^t)\right)\Big]_{y_i^t}$, where $[\cdot]_y$ denotes the $y$-th component. The surrounding text now explicitly identifies the target confidence as the softmax probability assigned by the model to the true label, removing the ambiguity in the original notation.

---

### Review · Reviewer_NA6H · 2026-04-13

**Summary Of Contributions:**

This paper presents a rehearsal-based algorithm for the continual supervised classification problem. The algorithm curates samples close to the decision boundary in a replay buffer, while maintaining equal allocation across tasks and target classes. Samples with highly variable predicted class during early epochs in each task are deemed challenging and stored in the learner’s episodic memory to help avoid catastrophic forgetting across tasks.

The authors perform a series of empirical evaluations, testing their algorithm on class-incremental and task-incremental settings, using CIFAR-100 and two variants of ImageNet datasets. Other experiments investigate run-time, alternative buffer update policies, and sensitivity to a hyperparameter of the proposed algorithm and support the authors claim that their algorithm improves continual learning performance while being computationally inexpensive.

Strengths
1. The paper is written clearly and is easy to read. The authors do a good job at situating their contributions within the existing literature. I found the related work section to be a nice summary of the rehearsal-based continual learning literature.
2. The method is well motivated and explained. The authors use clear diagrams and a motivating experiment (fig 3) to support the exposition. Although some of the claims in these experiments are not as evident as the text implies.
3. Experiment setup is described in sufficient detail. Recorded metrics are explained and justified. The experiments cover a large range of baselines.

Weaknesses
1. I think the main weakness of the paper is claiming the algorithm is applicable to a broader set of problems than it seems. For instance, the algorithm only works for classification problems where the learner is aware of clear task boundaries, and can iterate over each task’s data for several epochs before moving on to another task.
2. I think the proposed buffer update policy is sensitive to target noise. The variability in predicted targets may be due to noise rather than change in the decision-boundary, potentially causing noisy samples to be kept in the buffer harming performance, similar to the noisy TV problem. However this possibility is not discussed in the paper.
3. The authors run each experiment for 5 seeds and report mean performance, omitting any measure of variability or confidence. However the mean performance of several algorithms in many cases are close to each other and requires more information for evaluation.

**Audience:**

Yes

**Audience Explanation:**

This paper presents an algorithm for the continual supervised learning problem and I think a large set of individuals in TMLR's audience will be interested in reading this paper. The paper does a good job motivating the problem and proposed algorithm by giving a nice survey of current approaches for rehearsal-based continual supervised learning, which is also a valuable resource for the audience.

The following is a non-exhaustive list of who might be interested in reading this paper:
1. Those interested in rehearsal-based continual learning algorithms
2. Neural networks researchers interested in loss of plasticity, and catastrophic forgetting
2. Reinforcement learning researchers, particularly those interested in experience replay buffers

**Broader Impact Concerns:**

I do not see any ethical implications of this work that would require adding a Broader Impact Statement.

**Claims And Evidence:**

No

**Claims Explanation:**

I think at the moment two of the claims in the paper remain unconvincing.

1. Scope of the applicability of algorithm. The paper claims the proposed algorithm is applicable in "continual learning" problems, however it only works for a subset of continual learning problems (see weakness 1 and requested changes 2).
2. The experimental results do not report a measure of statistical confidence, which undermines the claims made when comparing performance of the algorithms (see weakness 3 and requested changes 1).

**Requested Changes:**

Important changes
1. The empirical results should include a measure of confidence such as bootstrap CI or other statistical tests to back the claims. Alternatively, the claims regarding superior performance of the proposed algorithm should be softened appropriately.
2. The paper should clearly describe the assumptions and scope of the applicability of the proposed algorithm. For example, the algorithm seems to be only applicable to classification problems with clear task boundaries.

Minor changes not impacting my decision
- Incorrect citation style. If the citation is not part of the sentence it should be enclosed in round brackets.
- Address the possibility that the proposed algorithm can be sensitive to noise.
- Explain how the hyperparameters were selected, both in the main text and the appendix.
- Explain the task incremental and class incremental problem settings to make the paper more self contained.
- Section 5 should have a preamble before the first subsection.
- Line 11 of the pseudocode suggests the optimal parameters are found in each step, which is not what the algorithm is doing. Instead this should be one step of SGD.

---

> ### Author Response · Authors · 2026-05-02
>
> **Response to reviewer concern on statistical confidence.**
> We thank the reviewer for raising this important concern, which has substantially improved the rigor of our empirical claims. We have made the following changes in the revised manuscript:
>
> (1) Standard deviations. Tables 2 and 3 (main results, CIL and TIL) now report mean $\pm$ standard deviation across the 5 seeds for every cell, allowing direct visual comparison of run-to-run variability across methods.
>
> (2) Paired statistical tests. We performed paired two-sided $t$-tests ($\alpha = 0.05$, $n=5$ matched seeds) comparing CGR against the strongest competitor, CSReL-CL-Prv, on every (dataset, buffer) cell, every per-dataset pooled comparison, and the overall 45-run pooled comparison, for both ACC and BWT under CIL and TIL. Results are reported in new Tables 11 and 12 in the appendix, and significant cells are marked with $\ast$ in Tables 2 and 3 directly. Pairing is on (dataset, buffer, seed), so each test contrasts two runs that differ only in method.
>
> (3) Softened claims. Following the reviewer's suggestion, we revised Sections 5.2 (Class Incremental Learning, Task Incremental Learning, Modularity, Update Policies) and the abstract to align our claims with what the tests actually support.
>
> **Response to reviewer concern on scope and applicability.**
> Thank you for raising this important point. We agree that the current manuscript describes the applicability of CGR too broadly. CGR is designed for supervised classification-based continual learning (CL) with identifiable task boundaries and multi-epoch access to the current task data. This is because the method computes the temporal variance of the model’s target-class confidence over the first $E$ epochs and performs buffer updates at task transitions. Therefore, the current formulation is directly applicable to task-incremental and class-incremental supervised classification settings, but not to strict single-pass online CL, task-free CL, regression, unsupervised CL, or reinforcement-learning settings without modification.
>
> In the revised manuscript, we have added a “Scope and Assumptions” paragraph in Section 4 and softened the abstract, introduction, and conclusion to avoid suggesting that CGR applies to all CL settings. We have stated that CGR is intended for supervised classification-based CL with clear task boundaries, matching the benchmarks evaluated in the paper. We have also discussed broader settings, such as task-free or online CL, as future work rather than as direct applications of the current method.
>
> **Response on citation style.** We thank the reviewer for catching this. We have replaced all parenthetical citations with \citep{...} throughout the manuscript, while keeping \citet{...} for the cases where the cited authors are part of the sentence's grammatical structure. The revision uses the correct citation style consistently.
>
> **Response on noise sensitivity and the "noisy TV" failure mode.** We thank the reviewer for naming this failure mode explicitly. We agree that it is a real concern for the variance-based selection criterion and have added a discussion in Section 5.3 addressing it directly. Our defense is mechanistic: (i) variance is measured only over the first $E=2$--$5$ epochs, before networks memorize noisy labels (Zhang et al., 2016; Arpit et al., 2017), so mislabeled samples remain in the low-variance "hard outlier" region during selection; CGR is designed to capture resolvable (transient) uncertainty rather than irreducible noise. (ii) Per-class buffer balance bounds the worst-case effect of a noisy class. We acknowledge in the same paragraph that we expect the early-epoch and class-balance design to provide partial robustness, but caution that under sufficiently high noise, the variance signal could degrade, and pairing CGR with an explicit noise-detection mechanism (e.g., loss-based filtering as in Arazo et al., 2019) is a natural extension.
>
> Chiyuan Zhang, Samy Bengio, Moritz Hardt, Benjamin Recht, and Oriol Vinyals. Understanding deep learning requires rethinking generalization. arXiv preprint arXiv:1611.03530, 2016.
>
> Devansh Arpit, Stanisław Jastrzębski, Nicolas Ballas, David Krueger, Emmanuel Bengio, Maxinder S Kanwal, Tegan Maharaj, Asja Fischer, Aaron Courville, Yoshua Bengio, et al. A closer look at memorization in deep networks. In International conference on machine learning, pp. 233–242. PMLR, 2017.
>
> Eric Arazo, Diego Ortego, Paul Albert, Noel O’Connor, and Kevin McGuinness. Unsupervised label noise modeling and loss correction. In International conference on machine learning, pp. 312–321. PMLR, 2019.

---

> ### Author Response · Authors · 2026-05-02
>
> **Response on hyperparameter selection.** We thank the reviewer for this point and have clarified the procedure in Section 5.1 ("Implementation Details" paragraph) and Appendix A.1. For all baselines, we use the hyperparameter values reported in the original papers and the Mammoth framework without further tuning, for fair comparison. For CGR, the only method-specific hyperparameter is $E$, which we searched over $\{2, \dots, 10\}$ per (dataset, buffer-size) pair, points to the sensitivity analysis already provided in Figure 3 (c.1, c.2), where CGR's performance is stable across $E \in [2, 6]$ on all three datasets.
>
> **Response on self-containedness of CIL and TIL.** We thank the reviewer for the suggestion. The revised manuscript adds a new paragraph at the end of Section 3 (Preliminaries), defining the task-incremental and class-incremental learning protocols. The paragraph clarifies that both settings share the same training procedure—a sequence of tasks with disjoint class sets—and differ only at evaluation: TIL provides the task identity at test time and restricts prediction to that task's class set, whereas CIL provides no task identity and requires prediction over all classes seen so far, making CIL the harder of the two. The corresponding sentence in Section 5.1 (“Metrics” paragraph) has been shortened to refer back to the new definition.
>
> **Response on Section 5 preamble.** We thank the reviewer for this suggestion. Section 5 now opens with a short preamble that previews the structure of the experimental evaluation: 5.1 covers the experimental setup, 5.2 reports main results, and 5.3 presents ablations.
>
> **Response on Algorithm 1, line 11.** We thank the reviewer for catching this notational error. The reviewer is correct that line 11, as written, suggests an exact minimization at each step, which is not what the algorithm performs. We have replaced it with a single SGD update, $\theta \leftarrow \theta - \eta\\nabla_\theta\\text{CrossEntropy}(f_\theta(x), y)$, and removed the $\theta_t^*$ notation from the inner loop.

---

> > ### Comment · Reviewer_NA6H · 2026-05-12
> >
> > Thank you for the detailed response and revising the paper, I like the changes.
> >
> > The updated manuscript addresses my original concerns and my answer to "Are the claims made in the submission supported by accurate, convincing and clear evidence" is now changed to yes!

---

### Decision · Action_Editor_MfEK · 2026-06-24

**Recommendation:** Reject

**Audience:**

Yes

**Audience Explanation:**

Identifying which samples to store for rehearsal is of broad interest, even beyond continual learning. This work is limited to classification, so it limits the audience, but there are potentially ways to develop a similar approach beyond classification.

**Claims And Evidence:**

No

**Claims Explanation:**

The reviewers have given useful feedback and the authors have improved the paper. There are, however, still several issues to resolve. I have outlined them below.

The papers is pitched too broadly. It is restricted to classification (and with other restrictions too like having task boundaries), and this should be made more clear (including in the title).

This paper builds on the confidence measures introduced by Swayamdipta et al.. Figure 3 is similar to the figure used in that work to motivate the measure. Swayamdipta et al. used it for a different purpose, essentially to make higher quality datasets, but it is a definitely a related goal. They also point out improvements in generalization, not just on compressing to the most useful samples. Section 4.1 should be moved earlier in the paper and connections to this important foundational work more clearly outlined. Additionally, though I do appreciate what was already added by the authors, there could still be more investigation into why this approach identifies boundary samples, either with more discussion or even some experiments testing how often this intuition is or is not violated. You could leverage any insights in the Swayamdipta paper (and any follow ups) more too.

The way the hyperparameters are chosen is not sufficiently clear. The actual values are outlined in the appendix, but they are different per setting and it is not explained where they came from. The choice of hyperparameters can have a big impact on results, so understanding how this is done is critical for interpreting the results and conclusions.

It is surprising, and could maybe use some more explanation, that the method already works with 2 epochs. I believe this means only 2 samples are used in the variance computation for confidence measure. This could be explained more.

The tables are very hard to read. I wonder if you can reconsider how you present these, or also consider providing some plots that highlight certain comparisons, like those relative to CSReL.


Finally, it would be important to understand how consistent these results are with more or less replay. There are currently 50 epochs, for example, per task. How different is this if there are more or less? It is ok if it does not help with more or less epochs, but I expect significant interaction with the number of epochs and it is important to understand the behavior when bringing this idea to new settings.

Minor comments:
1. I found it confusing that Gamma_e(i,t) was defined on x_i^t and y_i^t. I initially thought this was being computed for all possible y, not just the label observed. I think this would be more clear to use Gamma_e(i,t)  to be clear its for a specific datapoint.

2. it is hard to consider both ACC and BWT. I wonder if there is some way to have one number reflecting BWT only when ACC is reasonable.

**Resubmission Of Major Revision:**

The authors may consider submitting a major revision at a later time.